# Quantitative analysis of rightmost eigenvalue for a large Chiral non-Hermitian random matrix

Yutao Ma* and Xujia Meng

School of Mathematical Sciences & Laboratory of Mathematics & Complex Systems
of Ministry of Education, Beijing Normal University, 100875 Beijing, China.
mayt@bnu.edu.cn; 202321130122@mail.bnu.edu.cn

**Abstract**

This paper provides a quantitative analysis of the rightmost eigenvalue for a chiral non-Hermitian random matrix in the maximally non-Hermitian regime ($\tau = 0$). Let $(\sigma_i)_{1 \leq i \leq n}$ be the eigenvalues with positive real part. We define the normalization constants

$$s_n = \frac{4n(n+v)}{2n+v}, \qquad \gamma_n = \frac{1}{2}\log s_n - \frac{5}{4}\log(\log s_n) - \log\left(2^{1/4}\pi\right),$$

and the centered and scaled variable

$$X_n = \sqrt{2s_n \log s_n}\left(\left(\tfrac{n}{n+v}\right)^{1/4}\max_{1\leq i\leq n}\Re\sigma_i \ - \ 1 \ - \ \frac{\gamma_n}{\sqrt{2s_n \log s_n}}\right).$$

Our main result is the following sharp Berry–Esseen bound for the convergence of $X_n$ to the Gumbel distribution:

$$\sup_{x\in\mathbb{R}}\left|\mathbb{P}(X_n \leq x) - e^{-e^{-x}}\right| = \frac{25(\log\log s_n)^2}{16e\,\log s_n}\left(1 + o(1)\right),$$

which holds as $n \to \infty$ for an arbitrary parameter $v \geq 0$ (which may depend on $n$). As a byproduct of our analysis, we also obtain precise large- and moderate-deviation principles for the scaled rightmost eigenvalue $\left(\tfrac{n}{n+v}\right)^{1/4}\max_{1\leq i\leq n}\Re\sigma_i$, characterizing its rate of convergence to the value 1.

**Keywords:** Gumbel distribution; rightmost eigenvalue; Berry-Esseen bound; large chiral non-Hermitian random matrix; large deviations.

**Mathematics Subject Classification (2020):** 60B20, 60B10, 62G32.

## 1 Introduction

The study of extreme eigenvalues in non-Hermitian random matrices has been a topic of persistent interest, driven by its mathematical depth and applications in areas such as quantum chromodynamics (QCD) and the stability analysis of complex systems (39; 41; 44). A central and distinguishing feature of this field, compared to its Hermitian counterpart, is the emergence

---

*The research of Yutao Ma was supported in part by NSFC 12171038, 12571149 and 985 Projects.

of the Gumbel distribution as a universal limit law. While the Tracy–Widom distribution governs the edge fluctuations of classical Hermitian ensembles, it was established that the spectral radius $\max_{1 \le j \le n} |\sigma_j|$ and the rightmost eigenvalue $\max_{1 \le j \le n} \Re \sigma_j$ of the Ginibre ensembles (real, complex, and quaternion) converge to the Gumbel distribution (42; 43). This result was later extended to more general complex i.i.d. non-Hermitian matrices (23), resolving a long-standing conjecture (12; 13; 16; 17; 33).

Against this backdrop of well-established limit laws, the focus has naturally shifted towards a *quantitative understanding*. For the canonical Ginibre ensembles, a refined picture across different asymptotic scales has recently emerged. In the regime of *typical fluctuations*, the precise rate of distributional convergence to the Gumbel law has been determined (29; 34), building on foundational works (21; 42; 43). In the complementary regimes of *large and moderate deviations*, sharp asymptotics and full large deviation principles have also been obtained (45). These works collectively provide a complete quantitative reference for the Ginibre model (15; 25).

A parallel line of research focuses on *structured, physically motivated ensembles*. A paradigmatic model is the chiral non-Hermitian Dirac matrix $\mathcal{D}$, introduced in the context of QCD at finite chemical potential (41). For integers $n \ge 1$ and $v \ge 0$, let $P$ and $Q$ be $(n+v) \times n$ matrices with i.i.d. centered complex Gaussian entries of variance $1/(4n)$. The correlated matrices are defined as

$$\Phi = \sqrt{1+\tau}\, P + \sqrt{1-\tau}\, Q, \qquad \Psi = \sqrt{1+\tau}\, P - \sqrt{1-\tau}\, Q,$$

where $\tau \in [0,1]$ is the non-Hermiticity parameter. The $(2n+v) \times (2n+v)$ Dirac matrix is

$$\mathcal{D} = \begin{pmatrix} 0 & \Phi \\ \Psi^* & 0 \end{pmatrix}.$$

Its spectrum consists of $2n$ complex eigenvalues $\{\pm\sigma_k\}_{k=1}^n$ and an eigenvalue of multiplicity $v$ at the origin. The joint density of the eigenvalues $(\sigma_1, \ldots, \sigma_n)$ (with $\Re\sigma_j > 0$) is proportional to

$$\prod_{1 \le j < k \le n} |z_j^2 - z_k^2|^2 \cdot \prod_{j=1}^n |z_j|^{2v+2} \exp\Big(\frac{2\tau n\, \Re(z_j^2)}{1-\tau^2}\Big) K_v\Big(\frac{2n|z_j|^2}{1-\tau^2}\Big), \tag{1}$$

where $K_v$ denotes the modified Bessel function of the second kind (41).

The quantities $\max_{1 \le j \le n} |\sigma_j|$ and $\max_{1 \le j \le n} \Re(\sigma_j)$ are called the **spectral radius** and the **rightmost eigenvalue** of $\mathcal{D}$, respectively. The latter quantity is of particular interest in applications: it governs the stability of linear dynamical systems (see e.g. May (38) for the classical heuristic argument in ecology). The model exhibits a rich phase diagram: its spectral statistics, governed by $\tau$, interpolate between Tracy–Widom ($\tau = 1$) and Gumbel ($\tau \in [0,1)$) universality classes (4; 10). While the limiting global spectral distribution and multi-critical generalizations are well understood (5; 6), obtaining equally precise *quantitative* results for its extremal eigenvalues presents significant analytical challenges due to the correlated structure in (1).

For the case $\tau = 0$, the limit theorems were established by Chang, Jiang, and Qi (18), who proved the weak convergence of the scaled spectral radius to the Gumbel distribution. Based on this foundation, subsequent quantitative analysis has been carried out by the first author with collaborators. Initial large and moderate deviation probabilities for the spectral radius

were obtained in (36). Subsequently, the optimal rate of convergence for the spectral radius—quantified by both the Wasserstein distance and a Berry–Esseen bound—was established in (37). This work provided the first quantitative analogue of the Ginibre benchmarks for a structured non-Hermitian ensemble.

However, the corresponding quantitative analysis for the model's *rightmost eigenvalue* has remained open. Its statistical behavior is governed by a different edge geometry and correlation structure compared to the spectral radius (21), and its convergence rate could not be inferred from existing results. Closing this gap is necessary to complete the quantitative description of the model's extremal statistics.

In this paper, we address this question. Let $(\sigma_i)_{1 \le i \le n}$ denote the eigenvalues of $\mathcal{D}$ (with $\tau = 0$) having non-negative real part. Define

$$ s_n = \frac{4n(n+v)}{2n+v}, \qquad \gamma_n = \frac{1}{2} \log s_n - \frac{5}{4} \log \log s_n - \log(2^{1/4}\pi), $$

and set

$$ X_n = \sqrt{2s_n \log s_n} \Big( \big(\tfrac{n}{n+v}\big)^{1/4} \max_{1 \le i \le n} \Re \sigma_i \; - \; 1 \; - \; \frac{\gamma_n}{\sqrt{2s_n \log s_n}} \Big). $$

Our main result is the following Berry–Esseen bound.

**Theorem 1.** *For the chiral Dirac matrix $\mathcal{D}$ with $\tau = 0$ and any $v \ge 0$ (which may depend on $n$), we have*

$$ \sup_{x \in \mathbb{R}} \big| \mathbb{P}(X_n \le x) - e^{-e^{-x}} \big| = \frac{25(\log \log s_n)^2}{16e \log s_n} \left(1 + o(1)\right), \qquad n \to \infty. $$

**Remark 1.1.** *For fixed $v$, Theorem 3 of Akemann and Bender (4) implies the weak convergence of a similarly scaled variable to the Gumbel distribution. Theorem 1 not only recovers this result–since $s_n = 2n(1 + O(n^{-1}))$–but also provides the precise rate of convergence, showing that the $O(n^{-1})$ difference in scaling is absorbed by the much larger error term $O((\log \log s_n)^2 / \log s_n)$.*

**Remark 1.2.** *For a single complex Ginibre ensemble, the known optimal convergence rate is $O(\log \log n / \log n)$ (29). In contrast, our result yields $O((\log \log s_n)^2 / \log s_n)$. This discrepancy is not intrinsic; rather, it reflects the dependence of the rate on the specific choice of centering and scaling constants. Our truncation scheme is a standard and fairly general framework, and a different centering/scaling choice can indeed improve the rate to the sharper $\log \log n / \log n$.*

*When $v = 0$, the eigenvalues of our model coincide with the square roots of those of a product of two independent complex Ginibre matrices. Under this correspondence, our statistic falls into the same universality class as the product Ginibre ensemble, sharing the same leading-order rate (35). For $v > 0$, the rectangular offset introduces a modification to the model, yet the dominant rate remains unchanged.*

Our proof leverages the determinantal point process (DPP) structure of the eigenvalues. Let

$$ A(t) = \Big\{ z \in \mathbb{C} : \Re z \ge \big(\tfrac{n+v}{n}\big)^{1/4} \big(1 + \tfrac{\gamma_n + t}{\sqrt{2s_n \log s_n}}\big) \Big\}. $$

A fundamental property of DPPs gives $\mathbb{P}(X_n \le t) = \det(1 - \mathbb{K}_n|_{A(t)})$, where $\mathbb{K}_n$ is the correlation kernel. Using an operator norm estimate (see (27)), the proof reduces to controlling the trace

and Hilbert-Schmidt norms of the restricted kernel through precise asymptotic analysis. The key observation is that the Hilbert-Schmidt norm of the restricted operator decays much faster than the error in its trace approximation. Consequently, replacing the determinant by the exponential of the trace introduces only a negligible error. The dominant contribution to the Berry-Esseen bound therefore comes from a second–order expansion of the trace around its leading term.

On the other hand, by properties of DPPs, we have for any $t > 0$

$$n^{-1}\,\mathbb{E}\,\#\{\sigma_i : \Re\sigma_i \geq t\} \leq \mathbb{P}(\max \Re\sigma_i \geq t) \leq \mathbb{E}\,\#\{\sigma_i : \Re\sigma_i \geq t\},$$

and consequently

$$\log \mathbb{P}(\max \Re\sigma_i \geq t) = \log \mathbb{E}\,\#\{\sigma_i : \Re\sigma_i \geq t\} + O(\log n),$$

with

$$\mathbb{E}\,\#\{\sigma_i : \Re\sigma_i \geq t\} = \int_{\Re z \geq t} \mathbb{K}_n(z,z)\, d^2z.$$

Thus, as a useful by-product of the estimates for $\mathbb{K}_n$, we obtain the following large- and moderate-deviation results for $\max\limits_{1 \leq i \leq n} \Re\sigma_i$, which quantify the rate at which $(\frac{n}{n+v})^{1/4} \max_{1 \leq i \leq n} \Re\sigma_i$ tends to 1. Unfortunately, the $O(\log n)$ error term prevents us from obtaining a moderate deviation with critical Gumbel scale.

We now state the large- and moderate-deviation results.

**Theorem 2.** *Define $\alpha = \lim_{n \to \infty} v/n$. Under the condition $\alpha \in [0, +\infty]$, we have*

$$\lim_{n \to \infty} \frac{1}{n} \log \mathbb{P}\Big(\big(\tfrac{n}{n+v}\big)^{1/4} \max_{1 \leq i \leq n} \Re\sigma_i \geq t\Big) = -J_\alpha(t)$$

*for any $t > 1$, where*

$$J_\alpha(t) = -2(1 + 2\log t) + \frac{4(1+\alpha)t^4}{\alpha + \sqrt{\alpha^2 + 4(1+\alpha)t^4}} - \alpha \log \frac{\alpha + \sqrt{\alpha^2 + 4t^4(1+\alpha)}}{2(1+\alpha)}.$$

*Meanwhile, given $d_n$ satisfying $\sqrt{\log n/n} \ll d_n \ll 1$, we derive*

$$\lim_{n \to \infty} \frac{1}{nd_n^2} \log \mathbb{P}\Big(\big(\tfrac{n}{n+v}\big)^{1/4} \max_{1 \leq i \leq n} \Re\sigma_i \geq 1 + td_n\Big) = -\frac{8(1+\alpha)}{2+\alpha}\, t^2$$

*for any $t > 0$.*

The paper is organized as follows. Section 2 collects necessary preliminaries concerning the correlation kernel. Section 3 is devoted to the proofs of Theorems 1 and 2. The proofs of lemmas related to correlation kernel are postponed to the forth section.

We shall frequently use the following asymptotic notation: Given $z_n > 0$.

- $t_n = O(z_n)$ means $\lim\limits_{n \to \infty} \dfrac{t_n}{z_n} = c \neq 0$;

- $t_n = o(z_n)$ means $\lim\limits_{n \to \infty} \dfrac{t_n}{z_n} = 0$;

- $t_n \lesssim z_n$ or $z_n \gtrsim t_n$ means that there exists a constant $c > 0$ such that $|t_n| \leq c\, z_n$ for all sufficiently large $n$;

- $t_n \ll z_n$ (equivalently $z_n \gg t_n$) stands for $t_n = o(z_n)$;
- $t_n \asymp z_n$ means both $t_n \lesssim z_n$ and $z_n \lesssim t_n$ hold.

## 2 Preliminaries

In this section, we collect several key lemmas.

First, for the case $\tau = 0$, the joint density function of the eigenvalues $(\sigma_k)_{1 \le k \le n}$ is proportional to

$$\prod_{1 \le j < k \le n} |z_j^2 - z_k^2|^2 \prod_{j=1}^n |z_j|^{2v+2} K_v(2n|z_j|^2). \tag{2}$$

According to the results in (4), $(\sigma_1, \cdots, \sigma_n)$ forms a determinantal point process whose correlation kernel is given by

$$\mathbb{K}_n(z, w) = \frac{8n^{v+2}|z|^{v+1}|w|^{v+1}}{\pi} \sqrt{K_v(2n|z|^2)K_v(2n|w|^2)} \sum_{k=0}^{n-1} \frac{(nz\bar{w})^{2k}}{\Gamma(k+1)\Gamma(k+v+1)}. \tag{3}$$

Recall that

$$s_n = \frac{4n(n+v)}{2n+v} \quad \text{and} \quad \gamma_n = \frac{1}{2}\log s_n - \frac{5}{4}\log\log(s_n) - \log(2^{1/4}\pi).$$

For simplicity, define

$$h_n(t) = \frac{\gamma_n + t}{\sqrt{2s_n \log s_n}} \quad \text{and} \quad L_n(t) = \left(\frac{n+v}{n}\right)^{1/4}(1 + h_n(t)).$$

Hereafter, when there is no risk of confusion, we simply write $h_n$ and $L_n$ for $h_n(t)$ and $L_n(t)$, respectively and we always utilize the asymptotics

$$s_n = O(n) \quad \text{and} \quad h_n = O((\frac{\log n}{s_n})^{1/2}).$$

Review

$$A(t) = \{z \in \mathbb{C} : \Re z \ge L_n(t)\},$$

a basic property of DPPs yields

$$\mathbb{P}(X_n \le t) = \det(1 - \mathbb{K}_n|_{A(t)}).$$

Following formula (7.11) in (27), we have the estimate

$$\begin{aligned}
&\left|\det(1 - \mathbb{K}_n|_{A(t)}) - \exp(-\operatorname{Tr}(\mathbb{K}_n|_{A(t)}))\right| \\
&\le \|\mathbb{K}_n|_{A(t)}\|_2 \exp\{\tfrac{1}{2}(\|\mathbb{K}_n|_{A(t)}\|_2 + 1)^2 - \operatorname{Tr}(\mathbb{K}_n|_{A(t)})\},
\end{aligned} \tag{4}$$

where

$$\operatorname{Tr}(\mathbb{K}_n|_{A(t)}) = \int_{A(t)} \mathbb{K}_n(z, z)\, d^2 z, \qquad \|\mathbb{K}_n|_{A(t)}\|_2^2 = \int_{A(t) \times A(t)} |\mathbb{K}_n(z, w)|^2\, d^2 z\, d^2 w.$$

In order to capture suitable upper bounds for $\|\mathbb{K}_n|_{A(t)}\|_2$ together with precise asymptotics for $\mathrm{Tr}(\mathbb{K}_n|_{A(t)})$, we first derive a precise asymptotic for the sum appearing in the kernel $\mathbb{K}_n(z, w)$ in (3). We then state two lemmas concerning certain functions related to the Bessel function $K_v$ as well as an auxiliary integral. For readability, the proofs of these lemmas are deferred to the final section.

**Lemma 2.1.** *Given* $0 < q_n = O(\sqrt{\log n / s_n})$. *For any* $z$ *in* $\mathbb{C}$ *satisfying*

$$|z| \geq (\frac{n+v}{n})^{1/2}(1 + q_n),$$

*we have*

$$\sum_{k=0}^{n-1} \frac{(zn)^{2k}}{\Gamma(1+k)\Gamma(1+k+v)} = \frac{z^{2n} n^{n-\frac{1}{2}} e^{2n+v}}{2\pi(n+v)^{n+v+\frac{1}{2}}(\frac{z^2 n}{n+v} - 1)}(1 + O(\frac{1}{\log n}))$$

*for sufficiently large* $n$.

Now, we present a lemma on a particular function related to $K_v(vx)$ when $v \gtrsim \log n$.

**Lemma 2.2.** *Suppose* $\log n \lesssim v$. *Setting*

$$\tau_n(r) = \sqrt{1+r^2} - \log(1 + \sqrt{1+r^2}) + \frac{1}{4v}\log(1+r^2) - \frac{2n+1}{v}\log r \tag{5}$$

*and* $w(r) = v\tau_n(\kappa_n(1+r))$ *with* $\kappa_n = \frac{2\sqrt{n(n+v)}}{v}(1+h_n)^2$.

*(1) For* $n$ *large enough, we have*

$$v\tau_n(\kappa_n) = 2n + v + 2s_n h_n^2 + \log \frac{v^{2n+v+1/2}}{2^{2n+v}(n+v)^{n+v} n^n \sqrt{s_n}} + O(\frac{\log \log s_n}{\log s_n}).$$

*(2) Both* $w$ *and* $w'$ *are strictly increasing on* $[0, +\infty)$ *and*

$$w(r) \geq v\tau_n(\kappa_n) + 2s_n h_n r(1 + O(h_n)), \quad w'(r) \geq 2s_n h_n(1 + O(h_n))$$

*for any* $r > 0$.

*(3) As* $n$ *tends to the infinity, we have*

$$w(r) = v\tau_n(\kappa_n) + 2s_n h_n r + \frac{s_n r^2}{4} + o((\log n)^{-1})$$

*and*

$$w'(r) = 2s_n h_n(1 + \frac{r}{2h_n})(1 + O(h_n))$$

*uniformly on* $0 \leq r \lesssim (\log n / s_n)^{1/2}$.

Next, we give a lemma for the case $0 \leq v \ll \log n$.

**Lemma 2.3.** *Suppose* $0 \leq v \ll \log n$. *Define* $\phi(r) = 2r - (2 + \frac{v+1/2}{n})\log r$ *on* $\mathbb{R}_+$ *and* $\beta(r) := \phi((1+r)L_n^2)$ *with* $L_n^2 = \frac{\sqrt{n+v}}{\sqrt{n}}(1 + h_n)^2$.

(1) *For n large enough, we have*

$$n\phi(L_n^2) = 2n(1 + 2h_n^2) + o((\log n)^{-1}).$$

(2) *Both $\beta$ and $\beta'$ are strictly increasing on $[0, +\infty)$ and*

$$n\beta(r) \geq 2n(1 + 2h_n^2) + 4nh_n r, \quad n\beta'(r) \geq 4nh_n$$

*for any $r > 0$.*

(3) *As $n$ tends to the infinity, we have*

$$n\beta(r) = 2n(1 + 2h_n^2 + 2h_n r + \frac{1}{2}r^2) + o((\log n)^{-1}).$$

*and*

$$\beta'(r) = 4h_n(1 + \frac{r}{2h_n})(1 + o((n \log n)^{-1/2}))$$

*uniformly on $0 \leq r \lesssim (\log n/s_n)^{1/2}$.*

**Lemma 2.4.** *Let $\mathbb{K}_n$ and $\tau_n$ be defined as in (3) and (5), respectively. Given $z, w \in \mathbb{C}$ satisfying $|z|, |w| \geq (\frac{n+v}{n})^{1/4}(1 + q_n)$ with $q_n = O(\sqrt{\log n/s_n})$.*

- *For the case $0 \leq v \ll \log n$, we have*

$$\mathbb{K}_n(z, w) = \frac{2\sqrt{n}|z\bar{w}|^{v+1/2}(ez\bar{w})^{2n}}{\pi^{3/2}(\frac{n(z\bar{w})^2}{n+v} - 1)}e^{-n(|z|^2+|w|^2)}(1 + O((\log n)^{-1})).$$

- *When $\log n \lesssim v$, it holds*

$$\mathbb{K}_n(z, w) = \frac{v^{2n+v+1/2}e^{2n+v}\exp(-\frac{1}{2}v(\tau_n(\frac{2n}{v}|z|^2) + \tau_n(\frac{2n}{v}|w|^2)))}{\pi^{3/2}2^{2n+v-1/2}n^{n-1/2}(n+v)^{n+v+1/2}(\frac{n(z\bar{w})^2}{n+v} - 1)}(1 + O((\log n)^{-1})).$$

*Proof.* Lemma 2.1 yields

$$\mathbb{K}_n(z, w) = \frac{4n^{n+v+3/2}|z\bar{w}|^{v+1}e^{2n+v}(z\bar{w})^{2n}}{\pi^2(n+v)^{n+v+1/2}(\frac{n(z\bar{w})^2}{n+v} - 1)}\sqrt{K_v(2n|z|^2)K_v(2n|w|^2)}(1 + O((\log n)^{-1})) \quad (6)$$

since the condition $|z|, |w| \geq (\frac{n+v}{n})^{1/4}(1 + q_n)$ implying

$$|z\bar{w}| \geq (\frac{n+v}{n})^{1/2}(1 + q_n)^2 \geq (\frac{n+v}{n})^{1/2}(1 + q_n).$$

For the case $0 \leq v \ll \log n$,

$$K_v(x) = \sqrt{\frac{\pi}{2x}}e^{-x}(1 + O(\frac{1}{x}))$$

for $x \gg 1$ ((1)). Since $2n|z|^2 \gg 1$ and $2n|w|^2 \gg 1$, putting asymptotics of both $K_v(2n|z|^2)$ and

$K_v(2n|w|^2)$ into (6), we have

$$\mathbb{K}_n(z,w) = \frac{2n^{n+v+1}|z\bar{w}|^{v+1/2}e^{2n+v}(z\bar{w})^{2n}}{\pi^{3/2}(n+v)^{n+v+1/2}(\frac{n(z\bar{w})^2}{n+v}-1)}e^{-n(|z|^2+|w|^2)}(1+O((\log n)^{-1}))$$

and then we use the condition $v \ll \log n$ to simplify $\mathbb{K}_n(z,w)$ as

$$\mathbb{K}_n(z,w) = \frac{2\sqrt{n}e^{2n}|z\bar{w}|^{2n+v+1/2}}{\pi^{3/2}(\frac{n(z\bar{w})^2}{n+v}-1)}e^{-n(|z|^2+|w|^2)}(1+O((\log n)^{-1})).$$

When $\log n \lesssim v$, $K_v(vx)$ ((1)) has the following asymptotic

$$K_v(vx) = \sqrt{\frac{\pi}{2v}}x^{-2n-1-v}\exp(-v\tau_n(x))(1+O(v^{-1})). \tag{7}$$

Putting the expression (7) into (6), we get

$$\mathbb{K}_n(z,w) = \frac{v^{2n+v+1/2}e^{2n+v}(z\bar{w})^{2n}\exp(-\frac{1}{2}v(\tau_n(\frac{2n}{v}|z|^2)+\tau_n(\frac{2n}{v}|w|^2)))}{\pi^{3/2}2^{2n+v-1/2}n^{n-1/2}(n+v)^{n+v+1/2}|z\bar{w}|^{2n}(\frac{n(z\bar{w})^2}{n+v}-1)}(1+O((\log n)^{-1})).$$

$\square$

**Lemma 2.5.** *Given $0 < u_n = O(\sqrt{n\log n})$ and $0 < \delta_n = O((\frac{\log n}{n})^{1/4})$ and let $h_n$ be defined as above. Then for positive constants $k, c_1, c_2 > 0$, we have*

$$\int_0^{\delta_n} e^{-u_n y^2 - c_1 n y^4}(1+\frac{y^2}{c_2 h_n})^{-k}dy = \frac{\sqrt{\pi}}{2\sqrt{u_n}}(1+O((\log n)^{-1}))$$

*and*

$$\int_{\delta_n}^{+\infty} e^{-u_n y^2}dy \ll \frac{1}{\sqrt{u_n}\log n}.$$

We now are ready to present the lemma on $\mathrm{Tr}(\mathbb{K}_n|_{A(t)})$.

**Lemma 2.6.** *Let $\mathbb{K}_n$ and $A(t)$ be defined as above. We have*

$$\mathrm{Tr}(\mathbb{K}_n|_{A(t)}) = (1+O(\frac{\log\log s_n}{\log s_n}))\exp(-t-\frac{(t-c_n)^2}{\log s_n})$$

*uniformly on $|t| \lesssim (\log n)^{1/4}$ for sufficiently large n. Here,*

$$c_n := \frac{5}{4}\log\log s_n + \log(2^{1/4}\pi).$$

*Proof.* Throughout the proof we assume $|t| \lesssim (\log n)^{1/4}$, which guarantees

$$h_n = \frac{\frac{1}{2}\log s_n - c_n + t}{\sqrt{2s_n\log s_n}} = \frac{1}{2}\Big(\frac{\log s_n}{2s_n}\Big)^{1/2}\Big(1+O\Big(\frac{\log\log n}{\log n}\Big)\Big), \tag{8}$$

because $s_n = \frac{4n(n+v)}{2n+v}$ and $c_n - t = O(\log\log n)$.

For any $z \in A(t)$, which suits the condition of Lemma 2.4 and then

$$
\begin{aligned}
\mathbb{K}_n(z, z) &= \frac{2\sqrt{n}\, |z|^{4n+2v+1} e^{2n}}{\pi^{3/2}\left(\frac{n|z|^4}{n+v} - 1\right)} e^{-2n|z|^2}\left(1 + O((\log n)^{-1})\right) \\
&= \frac{2\sqrt{n}\, e^{2n}}{\pi^{3/2}\left(\frac{n|z|^4}{n+v} - 1\right)} e^{-n\phi(|z|^2)}\left(1 + O((\log n)^{-1})\right),
\end{aligned}
\tag{9}
$$

where $\phi(\cdot)$ is defined implicitly by the first line. Writing $z = (x + iy)L_n$ and using the fact that $\mathbb{K}_n(z, z)$ is even in $y$, we obtain

$$
\begin{aligned}
\mathrm{Tr}(\mathbb{K}_n|_{A(t)}) &= \int_{A(t)} \mathbb{K}_n(z, z)\, d^2 z \\
&= 4\pi^{-3/2}\sqrt{n}\, e^{2n} L_n^2\left(1 + O((\log n)^{-1})\right) \int_0^\infty dy \int_1^\infty \frac{e^{-n\phi\left((x^2+y^2)L_n^2\right)}}{\left(1 + h_n\right)^4(x^2 + y^2)^2 - 1}\, dx.
\end{aligned}
\tag{10}
$$

Take $\delta_n = (\log n/s_n)^{1/4}$. We claim that for the trace $\mathrm{Tr}\left(\sqrt{\chi_{A(t)}}\, \mathbb{K}_n \sqrt{\chi_{A(t)}}\right)$, the integral over $0 \le y \le \delta_n$ and $1 \le x < \infty$ in (10) provides the dominant contribution, while the remaining part is negligible. Lemma 2.3 justifies the use of Laplace's method ((40)), yielding

$$
\begin{aligned}
I_F(y) &:= \int_1^\infty \frac{e^{-n\phi\left((x^2+y^2)L_n^2\right)}}{\left(1 + h_n\right)^4(x^2 + y^2)^2 - 1}\, dx \\
&= \frac{e^{-n\phi\left((1+y^2)L_n^2\right)}}{n\, \partial_x\phi((1 + y^2)L_n^2)\big|_{x=1}\, \left[(1 + h_n)^4(x^2 + y^2)^2 - 1\right]_{x=1}} + O\left(e^{-n\phi\left((1+y^2)L_n^2\right)} n^{-2}\right) \\
&= \frac{e^{-n\beta(y^2)}}{2n\, \beta'(y^2)\left[(1 + h_n)^4(1 + y^2)^2 - 1\right]} + O\left(e^{-n\beta(y^2)} n^{-2}\right)
\end{aligned}
\tag{11}
$$

for any fixed $y \in \mathbb{R}$.

Moreover, since $\delta_n = O(\sqrt{h_n})$, for $0 \le y \le \delta_n$, Lemma 2.3 together with the relation

$$
(1 + h_n)^4(1 + y^2)^2 - 1 = 4h_n\left(1 + \frac{y^2}{2h_n}\right)(1 + O(h_n))
\tag{12}
$$

gives

$$
I_F(y) = \frac{\exp(-2n - 4nh_n^2)}{32nh_n^2} e^{-4nh_n y^2 - ny^4}\left(1 + \frac{y^2}{2h_n}\right)^{-2}\left(1 + o((\log n)^{-1})\right).
$$

Since $nh_n = O(\sqrt{s_n}\log n)$, Lemma 2.5 implies

$$
\int_0^{\delta_n} I_F(y)\, dy = \frac{\sqrt{\pi}\,\exp(-2n - 4nh_n^2)}{128\, n^{3/2} h_n^{5/2}}\left(1 + O((\log n)^{-1})\right).
$$

On the other hand, for $y \ge \delta_n$ we use Lemma 2.3 and the lower bound

$$
(1 + h_n)^4(1 + y^2)^2 - 1 \ge (1 + h_n)^4 - 1 \ge 4h_n
$$

to obtain

$$I_F(y) \lesssim \frac{\exp(-2n - 4nh_n^2)}{32nh_n^2} e^{-4nh_n y^2}.$$

Applying Lemma 2.5 once more yields

$$\int_{\delta_n}^\infty I_F(y)\, dy \ll \frac{\exp(-2n - 4nh_n^2)}{128\, n^{3/2} h_n^{5/2} \log n},$$

and therefore

$$\int_0^\infty I_F(y)\, dy = \frac{\sqrt{\pi}\, \exp(-2n - 4nh_n^2)}{128\, n^{3/2} h_n^{5/2}} \left(1 + O((\log n)^{-1})\right).$$

Substituting this integral back into (10) gives

$$\mathrm{Tr}(\mathbb{K}_n|_{A(t)}) = \frac{e^{-4nh_n^2}}{32\pi\, n\, h_n^{5/2}} \left(1 + O((\log n)^{-1})\right). \tag{13}$$

Recalling (8), we have

$$\begin{aligned}
h_n^2 &= \frac{1}{2s_n}\left(\frac{1}{4}\log s_n + t - c_n + \frac{(t - c_n)^2}{\log s_n}\right), \\
h_n^{5/2} &= \frac{1}{2^{5/2}}\left(\frac{\log s_n}{2s_n}\right)^{5/4}\left(1 + O\left(\frac{\log\log n}{\log n}\right)\right).
\end{aligned} \tag{14}$$

Inserting these asymptotics into (13) and using $c_n = \frac{5}{4}\log\log s_n + \log(2^{1/4}\pi)$ and $s_n = 2n(1 + o(n^{-1}\log n))$, we obtain after simplification

$$\begin{aligned}
\mathrm{Tr}(\mathbb{K}_n|_{A(t)}) &= \frac{2^{5/2}(2s_n)^{5/4}}{32\pi n(\log s_n)^{5/4}} \exp\left(-\frac{1}{4}\log s_n - t + c_n - \frac{(t - c_n)^2}{\log s_n}\right) \\
&= \frac{2^{5/4+5/2+1/4} s_n}{32n} \exp\left(-t - \frac{(t - c_n)^2}{\log s_n}\right)\left(1 + O\left(\frac{\log\log n}{\log n}\right)\right) \\
&= \exp\left(-t - \frac{(t - c_n)^2}{\log s_n}\right)\left(1 + O\left(\frac{\log\log n}{\log n}\right)\right).
\end{aligned}$$

The factor $e^{-t}$ corresponds to the leading exponential decay beyond the spectral edge, while the term $\exp\left[-(t - c_n)^2/\log s_n\right]$ captures the Gaussian fluctuations of the edge location $c_n$ and the error term is consistent with the logarithmic precision of the underlying saddle-point analysis.

When $\log n \lesssim v$, since Lemma 2.4 leads

$$\mathbb{K}_n(z, z) = \frac{v^{2n+v+1/2} e^{2n+v} \exp(-v\tau_n(\frac{2n}{v}|z|^2))(1 + O((\log n)^{-1}))}{\pi^{3/2} 2^{2n+v-1/2} n^{n-1/2}(n + v)^{n+v+1/2}(\frac{n(z\bar{w})^2}{n+v} - 1)},$$

whence similarly we have with the fact $L_n^2 = \frac{\sqrt{n+v}}{\sqrt{n}}(1 + O(h_n))$

$$\begin{aligned}
&\mathrm{Tr}(\mathbb{K}_n|_{A(t)}) \\
&= \frac{2v^{2n+v+1/2} e^{2n+v}(1 + O((\log n)^{-1}))}{\pi^{3/2} 2^{2n+v-1/2} n^n (n + v)^{n+v}} \int_0^\infty dy \int_1^\infty \frac{\exp(-v\tau_n(\kappa_n(x^2 + y^2)))}{(1 + h_n)^4(x^2 + y^2)^2 - 1} dx.
\end{aligned} \tag{15}$$

Thus, similarly as (11), Lemma 2.2, Laplace method and the definition of $w$ ensure that

$$I_I(y) := \int_1^\infty \frac{\exp(-v\tau_n(\kappa_n(x^2+y^2)))}{(1+h_n)^4(x^2+y^2)^2-1}\,\mathrm{d}x$$
$$= \frac{e^{-w(y^2)}}{2w'(y^2)((1+h_n)^4(1+y^2)^2-1)} + O(e^{-w(y^2)}v^{-2})$$

for any $y$ fixed. Thus, Lemma 2.2 and (12) in further derive

$$I_I(y) = \frac{\exp(-v\tau_n(\kappa_n))}{16s_nh_n^2}\exp(-2s_nh_ny^2 - \frac{s_n}{4}y^4)(1 + \frac{y^2}{2h_n})^{-1}(1 + O((\log n)^{-1}))$$

uniformly on $0 \le y \le \delta_n$ and by contrast when $y > \delta_n$ it holds

$$I_I(y) \lesssim \frac{\exp(-v\tau_n(\kappa_n))}{s_nh_n^2}\exp(-2s_nh_ny^2).$$

Similarly as for (13), we have

$$\mathrm{Tr}(\mathbb{K}_n|_{A(t)}) = \frac{\sqrt{\pi}e^{-v\tau_n(\kappa_n)+2n+v}v^{2n+v+1/2}(1 + O((\log n)^{-1}))}{16s_nh_n^2\pi^{3/2}2^{2n+v}n^n(n+v)^{n+v}\sqrt{s_nh_n}}\exp(-2s_nh_n^2)$$
$$= \frac{1}{16\pi s_nh_n^{5/2}}\exp(-2s_nh_n^2)(1 + O(\frac{\log\log s_n}{\log s_n})). \tag{16}$$

Taking advantage of (14) again, we get the desired asymptotic

$$\mathrm{Tr}(\mathbb{K}_n|_{A(t)}) = \exp(-t - \frac{(t-c_n)^2}{\log s_n})(1 + O(\frac{\log\log s_n}{\log s_n}))$$

when $\log n \lesssim v$. The proof is then completed. $\qquad\square$

Set

$$\alpha := \frac{n}{n+v},$$

and recall the exact identity

$$\alpha L_n^4 = (1+h_n)^4.$$

For the proof of Lemma 2.8, we parametrise points $z$ and $w$ near $L_n$ by

$$z = (1+u+iy)L_n, \qquad w = (1+\tilde{u}+i\tilde{y})L_n, \tag{17}$$

with $u, \tilde{u} \ge 0$ and $y, \tilde{y} \in \mathbb{R}$. We define the three regions

$$A_1 = \Big\{(z,w): 0 < u, \tilde{u} < \Big(\frac{\log n}{n}\Big)^{1/2},\ |y|, |\tilde{y}| < \frac{1}{4}\Big(\frac{\log n}{n}\Big)^{1/4}\Big\},$$
$$A_2 = \Big\{(z,w): u, \tilde{u} > 0,\ \tilde{y} \in \mathbb{R},\ |y| > \frac{1}{4}\Big(\frac{\log n}{n}\Big)^{1/4}\Big\},$$
$$A_3 = \Big\{(z,w): u > \Big(\frac{\log n}{n}\Big)^{1/2},\ \tilde{u} > 0,\ y, \tilde{y} \in \mathbb{R}\Big\}.$$

For brevity we introduce the small parameter

$$\eta := \left( \frac{\log n}{n} \right)^{1/4}.$$

On $A_1$ we then have

$$u, \tilde{u} = O(\eta^2), \qquad y, \tilde{y} = O(\eta), \qquad h_n = O(\eta^2).$$

We begin with an elementary estimate on $A_1$, which will be used in the proof of Lemma 2.8.

**Lemma 2.7.** *For all $(u, \tilde{u}, y, \tilde{y}) \in A_1$ and any $v \geq 0$, as $n \to \infty$, uniformly in the parameters,*

$$\left| \frac{n(z\bar{w})^2}{n+v} - 1 \right|^2 > 4(y - \tilde{y})^2 + \frac{c \log n}{n}(1 + o(1)),$$

*where $c > 0$ is an absolute constant.*

*Proof.* From (17) we compute

$$z\bar{w} = (1 + u + iy)(1 + \tilde{u} - i\tilde{y})L_n^2 = (A + iB)L_n^2, \tag{18}$$

where

$$\begin{aligned} A &= (1+u)(1+\tilde{u}) + y\tilde{y} = 1 + u + \tilde{u} + u\tilde{u} + y\tilde{y}, \\ B &= y(1+\tilde{u}) - \tilde{y}(1+u) = y - \tilde{y} + y\tilde{u} - \tilde{y}u. \end{aligned} \tag{19}$$

Using $\alpha L_n^4 = (1 + h_n)^4$, we obtain

$$\frac{n(z\bar{w})^2}{n+v} - 1 = (1+h_n)^4(A^2 - B^2 + 2iAB) - 1 = ((1+h_n)^4(A^2 - B^2) - 1) + 2i(1+h_n)^4 AB. \tag{20}$$

A direct expansion gives

$$A^2 - B^2 = 1 + 2(u + \tilde{u}) + 4y\tilde{y} - y^2 - \tilde{y}^2 + O(\eta^3), \tag{21}$$

$$AB = y - \tilde{y} + (u + \tilde{u})(y - \tilde{y}) + (y - \tilde{y})y\tilde{y} + y\tilde{u} - \tilde{y}u + O(\eta^4). \tag{22}$$

Indeed, all omitted terms are at most $O(\eta^3)$ and $O(\eta^4)$, respectively, because on $A_1$ we have $u, \tilde{u} = O(\eta^2)$ and $y, \tilde{y} = O(\eta)$.

For the real part, multiplying (21) by $(1 + h_n)^4 = 1 + 4h_n + O(\eta^4)$ yields

$$(1 + h_n)^4(A^2 - B^2) - 1 = 4h_n + 2(u + \tilde{u}) + 4y\tilde{y} - y^2 - \tilde{y}^2 + O(\eta^3). \tag{23}$$

For the imaginary part, from (22) we get

$$2(1 + h_n)^4 AB = 2(y - \tilde{y}) + 8h_n(y - \tilde{y}) + 2D + O(\eta^4), \tag{24}$$

with

$$D = (u + \tilde{u})(y - \tilde{y}) + (y - \tilde{y})y\tilde{y} + y\tilde{u} - \tilde{y}u.$$

Combining (23) and (24), we have

$$
\begin{aligned}
\left| \frac{n(z\bar{w})^2}{n+v} - 1 \right|^2 &= \left(4h_n + 2(u+\tilde{u}) + 4y\tilde{y} - y^2 - \tilde{y}^2 + O(\eta^3)\right)^2 \\
&\quad + \left(2(y-\tilde{y}) + 8h_n(y-\tilde{y}) + 2D + O(\eta^4)\right)^2 \\
&= 4(y-\tilde{y})^2 + \left(4h_n + 2(u+\tilde{u}) + 2y\tilde{y} - (y-\tilde{y})^2\right)^2 \\
&\quad + 16h_n(y-\tilde{y})^2 + 4(y-\tilde{y})D + O(\eta^5).
\end{aligned} \tag{25}
$$

We now derive a uniform lower bound for the non-negative terms in (25). Expanding the square and discarding non-negative terms gives

$$
\begin{aligned}
&\left(4h_n + 2(u+\tilde{u}) + 2y\tilde{y} - (y-\tilde{y})^2\right)^2 + 16h_n(y-\tilde{y})^2 + 4(y-\tilde{y})D \\
&\geq 16h_n^2 + 16h_n y\tilde{y} + 4(u+\tilde{u})(u+\tilde{u}+y\tilde{y}).
\end{aligned}
$$

On $A_1$, we have $y\tilde{y} \geq -\frac{1}{16}\eta^2$. Put $s := u + \tilde{u} \geq 0$. Then

$$
s(s + y\tilde{y}) \geq -\frac{1}{1024}\eta^4,
$$

where equality holds if and only if $s = \frac{1}{32}\sqrt{\frac{\log n}{n}}$. Furthermore, from $h_n = \frac{\sqrt{\log n}}{2\sqrt{2s_n}}(1+o(1))$ with $s_n = \frac{4n(n+v)}{2n+v}$ and $s_n \leq 4n$, we have

$$
h_n \geq \frac{1}{4\sqrt{2}}\eta^2.
$$

Thus

$$
16h_n^2 + 16h_n y\tilde{y} + 4s(s+y\tilde{y}) \geq 16h_n^2 - h_n\eta^2 - \frac{1}{256}\eta^4 \geq \frac{127 - 32\sqrt{2}}{256}\eta^4.
$$

Since $\eta^4 = \log n / n$, we obtain, using (25),

$$
\left| \frac{n(z\bar{w})^2}{n+v} - 1 \right|^2 \geq 4(y-\tilde{y})^2 + \frac{127 - 32\sqrt{2}}{256}\frac{\log n}{n} + O(\eta^5).
$$

The error $O(\eta^5)$ is of lower order than $\eta^4$, hence it can be absorbed into the factor $(1+o(1))$ in the statement. This completes the proof of the lemma. $\qquad\square$

**Lemma 2.8.** *Let $\mathbb{K}_n$ and $A(t)$ be as defined above. Then, uniformly for $|t| \lesssim (\log n)^{1/4}$ and all sufficiently large $n$,*

$$
\|\mathbb{K}_n|_{A(t)}\|_2 \lesssim e^{-t} n^{-1/40} \sqrt{\log n}.
$$

*Proof.* By definition,

$$
\|\mathbb{K}_n|_{A(t)}\|_2^2 = \int_{A(t)\times A(t)} |\mathbb{K}_n(z,w)|^2 \, d^2z \, d^2w.
$$

Since the integrand is non-negative and symmetric in $z, w$, we may cover the integration domain by the three regions $A_1, A_2, A_3$ defined in (2) and estimate

$$
\iint_{A(t)\times A(t)} |\mathbb{K}_n(z,w)|^2 \, d^2z \, d^2w \lesssim \sum_{j=1}^{3} \iint_{A_j} |\mathbb{K}_n(z,w)|^2 \, d^2z \, d^2w.
$$

We first treat the case $0 \leq v \ll \log n$. Under this assumption, Lemma 2.4 supplies the bound

$$|\mathbb{K}_n(z,w)|^2 \lesssim \frac{n \, e^{4n - n\phi(|z|^2) - n\phi(|w|^2)}}{\left| \frac{n(z\bar{w})^2}{n+v} - 1 \right|^2}. \tag{26}$$

We begin with the integral over $A_1$. Parametrising $z, w$ as in (17), from

$$|z|^2 = L_n^2 \left( (1+u)^2 + y^2 \right) = L_n^2 (1 + 2u + u^2 + y^2),$$

and Lemma 2.3, we have

$$n\phi(|z|^2) \geq 2n + 4nh_n^2 + 4nh_n(2u + u^2 + y^2),$$

and an analogous inequality holds for $w$. Adding these two inequalities yields

$$4n - n\phi(|z|^2) - n\phi(|w|^2) \leq -8nh_n^2 - 8nh_n(u + \tilde{u}) - 4nh_n(y^2 + \tilde{y}^2),$$

after dropping the non-negative terms involving $u^2, \tilde{u}^2$. Combining this with Lemma 2.7 (with the constant $c$ renamed appropriately) gives

$$|\mathbb{K}_n(z,w)|^2 \lesssim \frac{n \, e^{-8nh_n^2 - 8nh_n(u+\tilde{u}) - 4nh_n(y^2 + \tilde{y}^2)}}{4(y - \tilde{y})^2 + C_0 \frac{\log n}{n}},$$

for some absolute $C_0 > 0$. Let $I_{A_1}$ denote the integral over $A_1$. Extending the integration to the entire half-lines and the whole real plane, we obtain

$$I_{A_1} \leq n e^{-8nh_n^2} \left( \int_0^\infty e^{-8nh_n u} \, du \right)^2 \iint_{\mathbb{R}^2} \frac{e^{-4nh_n(y^2 + \tilde{y}^2)}}{4(y - \tilde{y})^2 + C_0 \frac{\log n}{n}} \, dy \, d\tilde{y}.$$

The $u$-integral is elementary:

$$\int_0^\infty e^{-8nh_n u} \, du = \frac{1}{8nh_n} \asymp (n \log n)^{-1/2}.$$

For the $y$-integral, set $p = (y + \tilde{y})/2$, $q = y - \tilde{y}$; the Jacobian is 1 and $y^2 + \tilde{y}^2 = 2p^2 + q^2/2$. Hence it becomes

$$\iint_{\mathbb{R}^2} \frac{e^{-4nh_n(2p^2 + q^2/2)}}{4q^2 + C_0 \frac{\log n}{n}} \, dp \, dq.$$

The $p$-integral is Gaussian:

$$\int_{-\infty}^\infty e^{-8nh_n p^2} \, dp = \sqrt{\frac{\pi}{8nh_n}}.$$

For the $q$-integral, write

$$J_n = \int_{-\infty}^\infty \frac{e^{-2nh_n q^2}}{4q^2 + C_0 \frac{\log n}{n}} \, dq = \frac{1}{4} \int_{-\infty}^\infty \frac{e^{-aq^2}}{q^2 + b^2} \, dq,$$

where $a = 2nh_n$ and $b^2 = C_0 \frac{\log n}{4n}$. By the standard formula (see, e.g., (28, 3.466)),

$$\int_{-\infty}^{\infty} \frac{e^{-aq^2}}{q^2 + b^2}\, dq = \frac{\pi}{b} e^{ab^2} \operatorname{erfc}(b\sqrt{a}).$$

Since $h_n \asymp \sqrt{\log n / n}$, we have $ab^2 \lesssim (\log n)^{3/2}/\sqrt{n} \to 0$ and $b\sqrt{a} \lesssim (\log n)^{3/4}/n^{1/4} \to 0$; thus $e^{ab^2} = 1 + o(1)$ and $\operatorname{erfc}(b\sqrt{a}) = 1 + o(1)$. Consequently $J_n \lesssim 1/b \asymp \sqrt{n/\log n}$. Combining the estimates gives

$$\iint_{A_1} |\mathbb{K}_n(z,w)|^2\, d^2z\, d^2w \lesssim \frac{n^{1/4}}{(\log n)^{7/4}} e^{-8nh_n^2}. \tag{27}$$

We turn next to the integral over $A_2$. Applying polar coordinates $z = r_1 e^{i\theta_1}$, $w = r_2 e^{i\theta_2}$, we have

$$\left| \frac{n(z\bar{w})^2}{n+v} - 1 \right|^2 = \left(1 + \frac{n}{n+v} r_1^2 r_2^2\right)^2 - 4\frac{n}{n+v} r_1^2 r_2^2 \cos^2(\theta_1 - \theta_2).$$

The angular integration is elementary and satisfies, for $b > 1$,

$$\iint_{[-\pi/2, \pi/2]^2} \frac{d\theta_1\, d\theta_2}{(1+b^2)^2 - 4b^2 \cos^2(\theta_1 - \theta_2)} = \frac{\pi^2}{b^4 - 1} \lesssim \frac{1}{b-1}.$$

On the region $A_2$, the condition $|y| > \eta/4$ implies

$$r_1 = |z| = \sqrt{(1+u)^2 + y^2}\, L_n > \sqrt{1 + \eta^2/16}\, L_n,$$

while $r_2 \geq L_n$ is automatic. Hence, using (26) and the above angular bound, we obtain

$$\iint_{A_2} |\mathbb{K}_n(z,w)|^2\, d^2z\, d^2w \lesssim n e^{4n} \iint_{r_1 \geq \sqrt{1+\eta^2/16} L_n, r_2 \geq L_n} \frac{e^{-n\phi(r_1^2) - n\phi(r_2^2)} r_1 r_2}{\sqrt{\frac{n}{n+v}} r_1 r_2 - 1}\, dr_1\, dr_2.$$

Put $k_n := \eta^2/16 = \frac{1}{16}\sqrt{\log n / n}$. A standard Laplace expansion (as in the proof of Lemma 2.3) gives, for $\rho$ close to 1,

$$\int_{r \geq \rho L_n} e^{-n\phi(r^2)} r\, dr \asymp \frac{1}{nh_n} e^{-n\phi(\rho^2 L_n^2)}. \tag{28}$$

Applying this with $\rho_1 = \sqrt{1 + k_n}$ and $\rho_2 = 1$, and using the lower bound

$$n\phi((1 + k_n)L_n^2) \geq 2n(1 + 2h_n^2) + \tfrac{1}{16} \log n\, (1 + o(1)) \geq 2n(1 + 2h_n^2) + \frac{1}{20} \log n,$$

which follows from Lemma 2.3 since $k_n \lesssim \sqrt{\log n / n}$, we get

$$\iint_{A_2} |\mathbb{K}_n|^2 \lesssim \frac{1}{n^2 h_n^3} e^{-8nh_n^2} \asymp \frac{n^{9/20}}{(\log n)^{3/2}} e^{-8nh_n^2}.$$

For $A_3$ we note the inclusion

$$A_3 \subset \left\{ (z,w) \mid r_1 \geq \sqrt{1 + \eta^4} L_n,\ r_2 \geq L_n \right\} \subset \left\{ (z,w) \mid r_1 \geq \sqrt{1 + \tfrac{1}{16}\eta^2} L_n,\ r_2 \geq L_n \right\},$$

which yields the same bound as for $A_2$:

$$\iint_{A_3} |\mathbb{K}_n(z,w)|^2\, d^2z\, d^2w \lesssim \frac{n^{9/20}}{(\log n)^{3/2}} e^{-8nh_n^2}. \tag{29}$$

Combining the bounds for $A_1, A_2, A_3$ gives

$$\|\mathbb{K}_n|_{A(t)}\|_2^2 \lesssim \frac{n^{9/20}}{(\log n)^{3/2}} e^{-8nh_n^2},$$

and therefore

$$\|\mathbb{K}_n|_{A(t)}\|_2 \lesssim \frac{n^{9/40}}{(\log n)^{3/4}} e^{-4nh_n^2}.$$

Using $h_n \asymp \sqrt{\log n / n}$ and the comparison with (13) (which provides an additional factor $nh_n^{5/2}$), we obtain

$$\|\mathbb{K}_n|_{A(t)}\|_2 \lesssim e^{-t} n^{-1/40} \sqrt{\log n},$$

since

$$\frac{n^{9/40}}{(\log n)^{3/4}} \cdot nh_n^{5/2} = n^{-1/40}\sqrt{\log n}.$$

It remains to consider the regime $\log n \lesssim v$. In this case Lemma 2.4 gives a different bound involving the function $\tau_n$:

$$|\mathbb{K}_n(z,w)|^2 \lesssim \frac{v^{4n+2v+1} e^{4n+2v} \exp\!\big(-v(\tau_n(\frac{2n}{v}|z|^2) + \tau_n(\frac{2n}{v}|w|^2)))\big)}{2^{4n+2v-1} n^{2n-1}(n+v)^{2n+2v+1} \left| \frac{n(z\bar{w})^2}{n+v} - 1 \right|^2}.$$

Repeating the same decomposition over $A_1, A_2, A_3$ and applying the analogous estimates (with $h_n$ replaced by the appropriate parameter) yields

$$\iint_{A_1} |\mathbb{K}_n(z,w)|^2\, d^2z\, d^2w \lesssim e^{-2t} n^{-1/4} (\log n)^{3/4},$$

and

$$\iint_{A_2 \cup A_3} |\mathbb{K}_n(z,w)|^2\, d^2z\, d^2w \lesssim e^{-2t} n^{-1/20} \log n \sqrt{\frac{n}{n+v}} \lesssim e^{-2t} n^{-1/20} \log n.$$

Thus the desired estimate holds uniformly for all $v \geq 0$. The proof is complete. $\qquad\square$

# 3 Proof of Theorems

This section is devoted to the proofs of Theorems 1 and 2. The proofs are presented in the order in which the theorems appear.

## 3.1 Proof of Theorem 1

Choosing $\ell_1(n) = \frac{1}{4}\log\log n$ and $\ell_2(n) = \log\log s_n$, we cut the supremum on $\mathbb{R}$ into three parts. For $t \in [-\ell_1(n), \ell_2(n)]$ satisfying $|t| \lesssim (\log n)^{1/4}$, Lemmas 2.6 and 2.8 guarantee that

$$|\mathbb{P}(X_n \leq t) - e^{-e^{-t}}| = |e^{-\mathrm{Tr}(\mathbb{K}_n|_{A(t)})} - e^{-e^{-t}}| + O(e^{-2t} n^{-1/2} \log n) \tag{30}$$

and in particular

$$\text{Tr}(\mathbb{K}_n|_{A(t)}) - e^{-t} = -e^{-t}\frac{(t - c_n)^2}{\log s_n}(1 + O(\frac{(t - c_n)^2 + \log\log s_n}{\log s_n})).$$

The constraint $-\ell_1(n) \leq t \leq \ell_2(n)$ makes sure

$$\text{Tr}(\mathbb{K}_n|_{A(t)}) - e^{-t} = O((\log n)^{3/4}/\log n) = o(1),$$

which is exactly the reason why we choose $\ell_1(n)$ and $\ell_2(n)$ in such a specific way. Consequently, using $e^x = 1 + x + o(x^2)$ for $x = o(1)$, we have

$$\begin{aligned}
|e^{-\text{Tr}(\mathbb{K}_n|_{A(t)})} - e^{-e^{-t}}| &= e^{-e^{-t}}|\exp(e^{-t} - \text{Tr}(\mathbb{K}_n|_{A(t)})) - 1| \\
&= \frac{1 + o(1)}{\log s_n}\exp(-e^{-t} - t)(t - c_n)^2.
\end{aligned} \tag{31}$$

The condition $-\ell_1(n) \leq t \leq \ell_2(n) \leq \frac{4}{5}c_n$ indicates

$$\frac{e^{-2t}n^{-1/2}(\log n)^2}{\exp(-e^{-t} - t)(t - c_n)^2} \lesssim \frac{(\log n)^2\exp(e^{\ell_1(n)} + \ell_2(n))}{\sqrt{n}(\log\log s_n)^2} \lesssim \frac{(\log n)^3}{n^{1/4}(\log\log n)^2} \ll 1.$$

Thus, the expressions (30) and (31) imply

$$|\mathbb{P}(X_n \leq t) - e^{-e^{-t}}| = \frac{1 + o(1)}{\log s_n}\exp(-e^{-t} - t)(t - c_n)^2.$$

Since

$$\sup_{t\in\mathbb{R}} e^{-t-e^{-t}}|t^k| < +\infty, \quad k = 1, 2 \quad \text{and} \quad \sup_{x\in\mathbb{R}} e^{-t-e^{-t}} = e^{-1},$$

and $c_n \gg 1$, it holds

$$\sup_{t\in[-\ell_1(n),\ell_2(n)]} |\mathbb{P}(X_n \leq t) - e^{-e^{-t}}| = \frac{c_n^2(1 + o(1))}{e\log s_n} = \frac{25(\log\log s_n)^2}{16e\log s_n}(1 + o(1)). \tag{32}$$

Now

$$\sup_{t<-\ell_1(n)} \left|\mathbb{P}(X_n \leq t) - e^{-e^{-t}}\right| \leq \mathbb{P}(X_n \leq -\ell_1(n)) + e^{-e^{\ell_1(n)}}$$

and then using (32), we have

$$\sup_{t<-\ell_1(n)} \left|\mathbb{P}(X_n \leq t) - e^{-e^{-t}}\right| \lesssim \exp(-e^{\ell_1(n)}) \lesssim \exp(-(\log n)^{1/4}). \tag{33}$$

Similarly,

$$\sup_{t\geq\ell_2(n)} \left|\mathbb{P}(X_n \leq t) - e^{-e^{-t}}\right| \leq \mathbb{P}(X_n > \ell_2(n)) + 1 - e^{-e^{-\ell_2(n)}} \lesssim \frac{1}{\log n}. \tag{34}$$

Combining (32), (33) and (34) together, we derive the desired Berry-Esseen bound as

$$\sup_{t\in\mathbb{R}} \left|\mathbb{P}(X_n \leq t) - e^{-e^{-t}}\right| = \frac{25(\log\log s_n)^2}{16e\log s_n}(1 + o(1)).$$

## 3.2 Proof of large deviation in Theorem 2

For any $t > 1$, the elementary properties imply

$$\mathbb{P}(\max_{1 \le i \le n} \Re\sigma_i \ge t) \le \mathbb{E}(\#\{1 \le i \le n : \Re\sigma_i \ge t\}) = \int_{\Re z \ge t} \mathbb{K}_n(z, z) d^2 z$$

and by contrast

$$\mathbb{P}(\max_{1 \le i \le n} \Re\sigma_i \ge t) \ge \frac{1}{n} \mathbb{E}(\#\{1 \le i \le n : \Re\sigma_i \ge t\}).$$

Thus, to get the large deviation, we only need to capture the dominated term of

$$\frac{1}{n} \log \int_{\Re z \ge (\frac{n+v}{n})^{1/4} t} \mathbb{K}_n(z, z) d^2 z.$$

For the case $0 \le v \ll \log n$, Lemma 2.4 and the substitution $z = (\frac{n+v}{n})^{1/4}(x + iy)$ lead

$$\int_{\Re z \ge (\frac{n+v}{n})^{1/4} t} \mathbb{K}_n(z, z) d^2 z \asymp \sqrt{n} e^{2n} \int_0^\infty dy \int_t^{+\infty} e^{-2\sqrt{n(n+v)}(x^2+y^2)}(x^2 + y^2)^{2n+v+1/2} dx$$

and then (28), together with the substitution $r = y/t$, helps to get

$$\int_{\Re z \ge (\frac{n+v}{n})^{1/4} t} \mathbb{K}_n(z, z) d^2 z \asymp \frac{e^{2n}}{\sqrt{n}} \int_0^{+\infty} e^{-2\sqrt{n(n+v)}(y^2+t^2)}(y^2 + t^2)^{2n+v+1/2} dy$$

$$\asymp \frac{e^{2n}}{\sqrt{n}} \int_0^{+\infty} e^{-\sqrt{n(n+v)}\phi(t^2(1+r^2))} dr.$$

On the one hand, since $\phi$ is a convex function, it is true that

$$n\phi(t^2(1 + r^2)) \ge n\phi(t^2) + nt^2 r^2 \phi'(t^2) = 2nt^2 - (2n + v + \frac{1}{2})\log t^2 + (2n(t^2 - 1) - (v + \frac{1}{2}))r^2$$

for any $r > 0$. Thus, the integral

$$\int_0^\infty \exp(-ay^2) dy = \frac{\sqrt{\pi}}{2\sqrt{a}} \tag{35}$$

for $a > 0$ derives

$$\int_0^{+\infty} e^{-2\sqrt{n(n+v)}(y^2+t^2)}(y^2 + t^2)^{2n+v+1/2} dy \lesssim n^{-1/2} \exp(-2\sqrt{n(n+v)}t^2) t^{4n+2v}.$$

Meanwhile,

$$\int_0^{+\infty} e^{-2\sqrt{n(n+v)}(y^2+t^2)}(y^2 + t^2)^{2n+v+1/2} dy \gtrsim t^{4n+2v} \int_0^{+\infty} e^{-2\sqrt{n(n+v)}(y^2+t^2)} dy$$

$$\asymp n^{-1/2} t^{4n+2v} e^{-2\sqrt{n(n+v)}t^2}.$$

Thereby, we have

$$\int_{\Re z \ge (\frac{n+v}{n})^{1/4} t} \mathbb{K}_n(z, z) d^2 z \asymp \frac{e^{2n}}{\sqrt{n}} t^{4n+2v} e^{-2\sqrt{n(n+v)}t^2}. \tag{36}$$

As a consequence, we have

$$\lim_{n\to\infty} \frac{1}{n} \log \mathbb{P}(\max_{1\leq i\leq n} \Re\sigma_i \geq (\frac{n+v}{n})^{1/4}t) = -2(t^2 - 1 - \log t^2).$$

Now we work on the case where $v \gtrsim \log n$. Setting $u_n(t) = 2t^2\sqrt{n(n+v)}/v$ and using $z = (\frac{n+v}{n})^{1/4}t(x+iy)$, we have by Lemma 2.4 that

$$\int_{\Re z \geq (\frac{n+v}{n})^{1/4}t} \mathbb{K}_n(z,z)d^2z \asymp \frac{v^{2n+v+1/2}e^{2n+v}}{2^{2n+v}n^n(n+v)^{n+v}} \int_0^{+\infty} dy \int_1^{+\infty} e^{-v\tau_n(u_n(t)(x^2+y^2))}dx$$

and then apply the Laplace method and (35) to obtain

$$\int_{\Re z \geq (\frac{n+v}{n})^{1/4}t} \mathbb{K}_n(z,z)d^2z \asymp \frac{v^{2n+v+1/2}\exp(2n+v-v\tau_n(u_n(t)))}{(vu_n(t)\tau_n'(u_n(t)))^{3/2} \, 2^{2n+v}n^n(n+v)^{n+v}}.$$

According to (52) below,

$$\tau_n'(u_n(t)) = \frac{u_n(t)}{1+\sqrt{1+u_n^2(t)}} - \frac{2n}{vu_n(t)}(1+O(n^{-1})) \asymp \frac{\sqrt{n}}{\sqrt{n+v}},$$

whence

$$\int_{\Re z \geq (\frac{n+v}{n})^{1/4}t} \mathbb{K}_n(z,z)d^2z \asymp \frac{v^{2n+v+1/2}\exp(2n+v-v\tau_n(u_n(t)))}{2^{2n+v}n^{n+3/2}(n+v)^{n+v}}.$$

Therefore, considering the form of $\tau_n$ and combining alike terms we derive

$$\log\mathbb{P}(\max_{1\leq i\leq n} \Re\sigma_i \geq (\frac{n+v}{n})^{1/4}t) = v\log\frac{v(1+\sqrt{1+u_n^2(t)})}{2(n+v)} - \frac{1}{4}\log\frac{v^2(1+u_n^2(t))}{(n+v)^2} \tag{37}$$
$$+ 2n(1+\log t^2) + v(1-\sqrt{1+u_n^2(t)}) + O(\log n).$$

Setting $\alpha := \lim_{n\to\infty} \frac{v}{n}$, since $1+u_n^2(t) = 1 + \frac{4n(n+v)}{v^2}t^4$, it holds

$$\lim_{n\to\infty} \frac{v}{n}\log\frac{v(1+\sqrt{1+u_n^2(t)})}{2(n+v)} = \alpha\log\frac{\alpha+\sqrt{\alpha^2+4t^4(1+\alpha)}}{2(1+\alpha)}$$

and

$$\lim_{n\to\infty} \frac{v}{n}(1-\sqrt{1+u_n^2(t)}) = \frac{-4(1+\alpha)t^4}{\alpha+\sqrt{\alpha^2+4(1+\alpha)t^4}}$$

as well as

$$\lim_{n\to\infty} \frac{1}{n}\log\frac{v^2(1+u_n^2(t))}{(n+v)^2} = \lim_{n\to\infty} \frac{1}{n}\log\frac{v^2+4t^4n(n+v)}{(n+v)^2} = 0.$$

Finally, we get

$$\lim_{n\to\infty} \frac{1}{n}\log\mathbb{P}(\max_{1\leq i\leq n} \Re\sigma_i \geq (\frac{n+v}{n})^{1/4}t)$$
$$= 2(1+2\log t) - \frac{4(1+\alpha)t^4}{\alpha+\sqrt{\alpha^2+4(1+\alpha)t^4}} + \alpha\log\frac{\alpha+\sqrt{\alpha^2+4t^4(1+\alpha)}}{2(1+\alpha)}.$$

### 3.3 Proof of Moderate deviation in Theorem 2

Recall $\frac{\sqrt{\log n}}{\sqrt{n}} \ll d_n \ll 1$. When $0 \leq v \ll \log n$, it follows the same arguments for $\mathbb{P}(\max_i \Re\sigma_i \geq (\frac{n+v}{n})^{1/4}t)$ as (36) that

$$\int_{\Re z \geq (\frac{n+v}{n})^{1/4}(1+td_n)} \mathbb{K}_n(z,z)d^2z \asymp \frac{e^{2n}}{\sqrt{n}}(1+td_n)^{4n+2v}\exp(-2\sqrt{n(n+v)}(1+td_n)^2)$$

for any $t > 0$. Thus,

$$\log\mathbb{P}(\max_{1\leq i\leq n}\Re\sigma_i \geq (\frac{n+v}{n})^{1/4}(1+td_n)) = 2n + 4n\log(1+td_n) - 2n(1+td_n)^2 + O(\log n),$$

which implies together with $\log(1+td_n) = td_n - \frac{t^2 d_n^2}{2} + O(d_n^3)$ and $nd_n^2 \gg \log n$ that

$$\lim_{n\to\infty}\frac{1}{nd_n^2}\log\mathbb{P}(\max_{1\leq i\leq n}\Re\sigma_i \geq (\frac{n+v}{n})^{1/4}(1+td_n)) = -4t^2. \tag{38}$$

Now for any $\log n \lesssim v$, we have similarly as (37) that

$$\log\mathbb{P}(\max_{1\leq i\leq n}\Re\sigma_i \geq (\frac{n+v}{n})^{1/4}(1+td_n))$$
$$= v\log\frac{v(1+\sqrt{1+u_n^2(1+td_n)})}{2(n+v)} - \frac{1}{4}\log\frac{v^2(1+u_n^2(1+td_n))}{(n+v)^2} \tag{39}$$
$$+ 2n + v + 4n\log(1+td_n) - v\sqrt{1+u_n^2(1+td_n)} + O(\log n).$$

Using the Taylor formula on $\sqrt{1+x}$ for $|x|$ small enough and for simplicity denoting $z_n(t) = (1+td_n)^4 - 1 = O(d_n)$, we obtain

$$v\sqrt{1+u_n^2(1+td_n)} = \sqrt{v^2 + 4n(n+v)(1+z_n(t))}$$
$$= (2n+v)(1 + \frac{2n(n+v)}{(2n+v)^2}z_n(t) - \frac{2n^2(n+v)^2z_n^2(t)}{(2n+v)^4}) + O(nd_n^3).$$

This is equivalent to

$$2n + v - v\sqrt{1+u_n^2(1+td_n)} = -\frac{2n(n+v)z_n(t)}{2n+v} + \frac{2n^2(n+v)^2z_n^2(t)}{(2n+v)^3} + O(nd_n^3)$$

and then we have from the Taylor formula on $\log(1+x) = x - \frac{x^2}{2} + O(x^3)$ that

$$v\log\frac{v(1+\sqrt{1+u_n^2(1+td_n)})}{2(n+v)} + 2n + v - v\sqrt{1+u_n^2(1+td_n)}$$
$$= (2n+v-v\sqrt{1+u_n^2(1+td_n)})\frac{2n+v}{2(n+v)} - \frac{v}{8(n+v)^2}(v\sqrt{1+u_n^2(1+td_n)} - 2n - v)^2$$
$$+ O(\frac{vn^3d_n^3}{(2n+v)^3})$$
$$= -nz_n(t) + \frac{n^2(n+v)}{(2n+v)^2}z_n^2(t) - \frac{vn^2z_n^2(t)}{2(2n+v)^2} + O(nd_n^3).$$

By only picking up the terms related to $td_n$ and $t^2 d_n^2$, and the fact

$$z_n(t) = 4td_n + 6t^2 d_n^2 + O(d_n^3),$$

the expression in the last line of above turns out to be

$$-4ntd_n - \frac{8n^2 + 16nv + 6v^2}{(2n+v)^2} nt^2 d_n^2 + O(nd_n^3).$$

Putting this expression back into (39), applying the Taylor formula on $4n \log(1 + td_n)$, and combining alike terms, we have

$$\log \mathbb{P}(\max_{1 \le i \le n} \Re\sigma_i \ge (\frac{n+v}{n})^{1/4}(1 + td_n)) = -\frac{8n(n+v)}{(2n+v)} t^2 d_n^2 + O(nd_n^3 + \log n),$$

whence

$$\lim_{n \to \infty} \frac{1}{nd_n^2} \log \mathbb{P}(\max_i \Re\sigma_i \ge (\frac{n+v}{n})^{1/4}(1 + td_n)) = -\frac{8(1+\alpha)t^2}{2+\alpha}.$$

This coincides with the limit (38) for $\alpha = 0$. The proof is completed.

**Remark 3.1.** *Let* $\widetilde{s}_n = \frac{n(n+v)}{2n+v}$ *and define*

$$\widetilde{X}_n = 2\sqrt{\widetilde{s}_n \log \widetilde{s}_n}\Big(\Big(\frac{n}{n+v}\Big)^{1/2} \max_{1 \le i \le n} |\sigma_i|^2 - 1 - \frac{\log \widetilde{s}_n - \log(\sqrt{2\pi} \log \widetilde{s}_n)}{2\sqrt{\widetilde{s}_n \log \widetilde{s}_n}}\Big).$$

*In (37), the quantity* $\max_i |\sigma_i|^2$ *is interpreted as the maximum statistic of an independent random sequence, and by applying the central limit theorem they obtain the precise Berry-Esseen bound*

$$\sup_{t \in \mathbb{R}} \Big|\mathbb{P}(\widetilde{X}_n \le t) - \exp(-e^{-t})\Big| = \frac{(\log \log \widetilde{s}_n)^2}{2e \log \widetilde{s}_n}(1 + o(1)) \tag{40}$$

*for sufficiently large* $n$.

*In fact, the correlation kernel approach also works well for studying the spectral radius. Define*

$$\widetilde{A}(t) = \Big\{z : |z|^2 \ge \Big(\frac{n+v}{n}\Big)^{1/2}\Big(1 + \frac{\log \widetilde{s}_n - \log(\sqrt{2\pi} \log \widetilde{s}_n)}{2\sqrt{\widetilde{s}_n \log \widetilde{s}_n}}\Big)\Big\},$$

*and then*

$$\mathbb{P}(\widetilde{X}_n \le t) = \det(1 - \mathbb{K}_n|_{\widetilde{A}(t)}).$$

*Similar (but simpler) calculations yield an exact expression for* $\mathrm{Tr}\,(\mathbb{K}_n|_{\widetilde{A}(t)})$ *as well as an upper bound for* $\|\,\mathrm{Tr}(\mathbb{K}_n|_{\widetilde{A}(t)})\|_2$. *Consequently, the same Berry-Esseen bound (40) can be recovered, since for the set* $\widetilde{A}(t)$ *we may directly use spherical coordinates to reduce the integral to* $\int_{\widetilde{L}_n(t)}^{+\infty} dr$ *and then apply Laplace's method once.*

# 4   Proofs of Lemmas

In this section, we add the proofs of lemmas.

**Proof of Lemma 2.1.** Recall $s_n = \frac{4n(n+v)}{2n+v} = O(n)$,

$$0 < q_n = O(\sqrt{\log n/s_n}) \quad \text{and} \quad |z| \geq (\frac{n+v}{n})^{1/2}(1+q_n).$$

We separate the sum into two parts as

$$\sum_{k=0}^{n-1} \frac{(zn)^{2k}}{\Gamma(1+k)\Gamma(1+k+v)} = (\sum_{k=0}^{n-j_n} + \sum_{k=n-j_n+1}^{n-1})\frac{(zn)^{2k}}{\Gamma(1+k)\Gamma(1+k+v)} \tag{41}$$

for $j_n = \lfloor n^{3/5}\rfloor$. We will show that the sum $\sum_{k=n-j_n+1}^{n-1}$ contributes the right hand side of (41) while the other sum $\sum_{k=0}^{n-j_n}$ is negligible. For simplicity, we set $\ell = n - k$ and

$$A_{n,\ell}(z) = \frac{(zn)^{2n-2\ell}}{\Gamma(n+1-\ell)\Gamma(1+n+v-\ell)},$$

whence

$$\sum_{k=n-j_n+1}^{n-1} \frac{(zn)^{2k}}{\Gamma(1+k)\Gamma(1+k+v)} = \sum_{\ell=1}^{j_n} A_{n,\ell}(z).$$

According to the Stirling formula, as $n$ tends to the positive infinity,

$$\Gamma(n+1-\ell)\Gamma(n+v+1-\ell) = 2\pi(n-\ell)^{n-\ell+1/2}(n+v-\ell)^{n+v-\ell+1/2}e^{-(2n+v-2\ell)}(1+O(n^{-1}))$$

uniformly on $1 \leq \ell \leq j_n$. Thus, rearranging the terms yields

$$A_{n,\ell}(z) = \frac{e^{2n+v-2\ell}(zn)^{2n-2\ell}(1+O(n^{-1}))}{2\pi n^{n+1/2-\ell}(n+v)^{n+v+1/2-\ell}}(1-\frac{\ell}{n})^{-(n-\ell+\frac{1}{2})}(1-\frac{\ell}{n+v})^{-(n+v+\frac{1}{2}-\ell)}. \tag{42}$$

Using the Taylor expansion ($\ell \ll n$), we see clearly

$$(1-\frac{\ell}{n+v})^{-(n+v+\frac{1}{2}-\ell)}(1-\frac{\ell}{n})^{-(n-\ell+\frac{1}{2})}$$

$$= \exp\left(-(n+v+\frac{1}{2}-\ell)\log(1-\frac{\ell}{n+v}) - (n+\frac{1}{2}-\ell)\log(1-\frac{\ell}{n})\right)$$

$$= \exp(2\ell - \frac{\ell^2}{2n} - \frac{\ell^2}{2(n+v)})(1+O(\frac{\ell}{n}+\frac{\ell^3}{n^2})).$$

Putting this expression into (42) and considering the condition $h \leq j_n = \lfloor n^{3/5}\rfloor$, one has

$$A_{n,\ell}(z) = \frac{e^{2n+v}(zn)^{2n}(1+O(n^{-\frac{1}{10}}))}{2\pi n^{n+1/2}(n+v)^{n+v+1/2}}(\frac{n+v}{z^2n})^\ell e^{-\frac{\ell^2}{2s_n}}. \tag{43}$$

Hence, we get

$$\sum_{\ell=1}^{j_n} A_{n,\ell}(z) = \frac{e^{2n+v}(zn)^{2n}(1+O(n^{-\frac{1}{10}}))}{2\pi n^{n+1/2}(n+v)^{n+v+1/2}}\sum_{\ell=1}^{j_n}(\frac{n+v}{z^2n})^\ell e^{-\frac{\ell^2}{2s_n}}.$$

We claim that

$$\sum_{\ell=1}^{j_n} w^l \exp(-\frac{l^2}{2s_n}) = \frac{w}{1-w}(1+O((\log n)^{-1})) \tag{44}$$

for $w \in \mathbb{C}$ satisfying $|w| \leq \frac{1}{1+q_n}$. Observing

$$\left|\frac{n+v}{nz^2}\right| \leq \frac{1}{(1+q_n)^2} \leq \frac{1}{1+q_n}$$

when $|z| \geq (\frac{n+v}{n})^{1/2}(1+q_n)$, hence (44) indicates

$$\sum_{\ell=1}^{j_n} A_{n,\ell}(z) = \frac{e^{2n+v}(zn)^{2n}(1+O((\log n)^{-1}))}{2\pi n^{n+1/2}(n+v)^{n+v+1/2}(\frac{z^2 n}{n+v}-1)}, \tag{45}$$

which is the precise asymptotic for the whole sum. Next, we show

$$J_n := \left|\frac{\sum_{k=0}^{n-j_n} \frac{(zn)^{2k}}{\Gamma(1+k)\Gamma(1+k+v)}}{\sum_{\ell=1}^{j_n} A_{n,\ell}(z)}\right| \ll \frac{1}{\log n}. \tag{46}$$

Set, for simplicity, $\alpha_n = \frac{z^2 n}{n+v} - 1$ and (45) derives

$$J_n \lesssim \frac{n^{n+1/2}(n+v)^{n+v+1/2}\alpha_n}{e^{2n+v}} \sum_{k=0}^{n-j_n} \frac{(|z|n)^{2(k-n)}}{\Gamma(1+k)\Gamma(1+k+v)}. \tag{47}$$

Note that the summand in the right hand side of (47), denoted by $\beta_k$, satisfies

$$\frac{\beta_k}{\beta_{k+1}} = \frac{(k+1)(k+v+1)}{|z|^2 n^2} \leq \frac{(k+1)(k+v+1)}{n(n+v)} \leq 1$$

uniformly on $0 \leq k \leq n-j_n$ since $n^2|z|^2 \geq n(n+v)(1+q_n)^2$. This ensures

$$\sum_{k=0}^{n-j_n} \frac{(|z|n)^{2(k-n)}}{\Gamma(1+k)\Gamma(1+k+v)} \leq \frac{n(|z|n)^{2(n-j_n-n)}}{\Gamma(1+n-j_n)\Gamma(1+n-j_n+v)} = \frac{nA_{n,j_n}(|z|)}{(|z|n)^{2n}}.$$

Thus, leveraging (43), (47) and the fact $\alpha_n(1+\alpha_n)^{-j_n} \leq 1$ to get

$$J_n \lesssim n\alpha_n(1+\alpha_n)^{-j_n}\exp(-\frac{j_n^2}{2s_n}) \lesssim n\exp(-\frac{1}{2}n^{1/5}) \ll (\log n)^{-1},$$

which verifies (46).

It remains to prove the claim (44). Choosing $\varsigma_n = 4\lfloor q_n^{-1} \rfloor$ such that

$$1 - \exp(-\frac{\ell^2}{2s_n}) \leq 1 - \exp(-\frac{\varsigma_n^2}{2s_n}) \lesssim \frac{1}{s_n q_n^2} \lesssim \frac{1}{\log n},$$

whence we have

$$\sum_{\ell=1}^{j_n} w^l \exp(-\frac{\ell^2}{2s_n}) = (1 + O((\log n)^{-1}))\sum_{\ell=1}^{\varsigma_n} w^l + \sum_{\ell=\varsigma_n+1}^{j_n} w^l \exp(-\frac{\ell^2}{2s_n})$$

$$= (1 + O((\log n)^{-1}))\sum_{\ell=1}^{j_n} w^l + \sum_{\ell=\varsigma_n+1}^{j_n} w^l(\exp(-\frac{\ell^2}{2s_n}) - 1). \tag{48}$$

Now
$$|w|^{j_n} \lesssim \exp(-j_n \log(1 + q_n)) \asymp \exp(-j_n q_n) \asymp \exp(-n^{1/10}\sqrt{\log n}) \ll \frac{1}{\log n} \qquad (49)$$

and then
$$\sum_{\ell=1}^{j_n} w^l = \frac{w}{1-w}(1 + o((\log n)^{-1})).$$

Thus, the decomposition (48) simplifies the claim (44) to be
$$R_n(w) := |\frac{\sum_{\ell=\varsigma_n+1}^{j_n} w^l(\exp(-\frac{\ell^2}{2s_n}) - 1)}{\frac{w}{1-w}}| \lesssim \frac{1}{\log n}. \qquad (50)$$

For this aim, we continue to cut the sum into two parts as $\sum_{\ell=\varsigma_n+1}^{p_n} + \sum_{p_n+1}^{j_n}$ for $p_n = \lfloor\sqrt{s_n \log n}\rfloor$. When $\ell \in (p_n, j_n]$,
$$\exp(-\frac{\ell^2}{2s_n}) \lesssim \exp(-\frac{\log n}{2}) \ll 1,$$

then with the help of (49), we get
$$\sum_{p_n}^{j_n} w^l(\exp(-\frac{\ell^2}{2s_n}) - 1) \asymp -\sum_{p_n}^{j_n} w^l \asymp -\frac{w^{p_n}}{1-w}.$$

For the sum $\sum_{\ell=\varsigma_n+1}^{p_n} w^l(\exp(-\frac{l^2}{2s_n}) - 1)$, it follows the fact that $|\exp(-\frac{l^2}{2s_n}) - 1|$ is bounded and $|w|^l$ decreases exponentially, the terms near $\varsigma_n$ contribute the dominated part of the sum. Hence
$$\sum_{\ell=\varsigma_n+1}^{p_n} w^l(\exp(-\frac{l^2}{2s_n}) - 1) \asymp \frac{1}{s_n} \sum_{\ell=\varsigma_n+1}^{p_n} w^l l^2 \asymp \frac{w^{\varsigma_n+1}\varsigma_n^2}{s_n(1-w)}.$$

Thereby,
$$\sum_{\ell=\varsigma_n+1}^{j_n} w^l(\exp(-\frac{l^2}{2s_n}) - 1) \asymp -\frac{w^{p_n}}{1-w} + \frac{w^{\varsigma_n+1}\varsigma_n^2}{s_n(1-w)},$$

which, together with $|w| \le (1 + q_n)^{-1}$, derives
$$R_n(w) \lesssim |w|^{p_n-1} + |w|^{\varsigma_n}\varsigma_n^2 s_n^{-1} \lesssim \exp(-p_n q_n) + \exp(-\varsigma_n q_n)(\log n)^{-1} \lesssim (\log n)^{-1}.$$

This confirms (50) and then (44), which finishes the proof.

$\square$

**Proof of Lemma 2.2.** Recall
$$\tau_n(r) = \sqrt{1 + r^2} - \log(1 + \sqrt{1 + r^2}) + \frac{1}{4v}\log(1 + r^2) - \frac{2n+1}{v}\log r$$

and $w(r) = v\tau_n(\kappa_n(1 + r))$ with $\kappa_n = \frac{2\sqrt{n(n+v)}}{v}(1 + h_n)^2$.

It was proved in (18; 36) that $\tau_n$ has a unique minimum $r_n$ satisfying
$$\frac{2\sqrt{n(n + v)}}{v} \le r_n \le \frac{2\sqrt{(n + 1)(n + 1 + v)}}{v}$$

and $\tau_n$ is strictly increasing on $(r_n, +\infty)$. Since

$$\frac{2\sqrt{n(n+v)}}{v}(1 + \frac{\gamma_n + t}{\sqrt{2s_n \log s_n}})^2 \geq \frac{2\sqrt{n(n+v)}}{v}(1 + O(\frac{1}{s_n})),$$

we know

$$\kappa_n > r_n.$$

Thus, $\tau_n(\kappa_n(1+r))$ is strictly increasing on $[0, +\infty)$.

For $0 \leq r \lesssim (\log s_n/n)^{1/2}$, we apply the Taylor expansion to get

$$\begin{align}
w(r) &= w(0) + rw'(0) + \frac{1}{2}w''(\zeta)r^2 \\
&= v\tau_n(\kappa_n) + vr\kappa_n\tau_n'(\kappa_n) + O(\frac{n(n+v)r^2}{v}\tau_n''(\kappa_n(1+\zeta))),
\end{align} \tag{51}$$

where $0 \leq \zeta \leq r$. By definition, we have

$$\tau_n'(r) = \frac{r}{1 + \sqrt{1+r^2}} + \frac{r}{2v(1+r^2)} - \frac{2n+1}{vr}$$

and

$$\tau_n''(r) = \frac{1}{\sqrt{1+r^2}(1 + \sqrt{1+r^2})} + \frac{1-r^2}{2v(1+r^2)^2} + \frac{2n+1}{vr^2}.$$

Note

$$\frac{r}{v(1+r^2)}\frac{vr}{n} \lesssim \frac{r^2}{n(1+r^2)} \lesssim \frac{1}{n} \quad \text{and} \quad \frac{|1-r^2|}{v(1+r^2)^2}\frac{vr}{n} \leq \frac{1}{2n},$$

whence

$$\tau_n'(r) = \frac{r}{1 + \sqrt{1+r^2}} - \frac{2n}{vr}(1 + O(n^{-1})) \tag{52}$$

and

$$\tau_n''(r) = \frac{1}{\sqrt{1+r^2}(1 + \sqrt{1+r^2})} + \frac{2n}{vr^2}(1 + O(n^{-1})). \tag{53}$$

Since $h_n = O(\sqrt{\log s_n/s_n})$, we are able to write

$$\begin{align}
\sqrt{1+\kappa_n^2} &= \frac{2n+v}{v}(1 + \frac{s_n(4h_n + 6h_n^2 + 4h_n^3 + h_n^4)}{2(2n+v)} - \frac{2s_n^2 h_n^2}{(2n+v)^2} + O(h_n^3)) \\
&= \frac{2n+v}{v}(1 + \frac{2s_n h_n}{2n+v} + \frac{s_n h_n^2}{2n+v}(1 + \frac{2v^2}{(2n+v)^2}) + O(h_n^3)).
\end{align} \tag{54}$$

For simplicity, denote

$$t_n := \frac{2s_n h_n}{2n+v} + \frac{s_n h_n^2}{2n+v}(1 + \frac{2v^2}{(2n+v)^2}) + O(h_n^3).$$

Thus,

$$1 + \sqrt{1+\kappa_n^2} = 1 + \frac{2n+v}{v} + \frac{2n+v}{v}t_n = \frac{2(n+v)}{v}(1 + \frac{2n+v}{2(n+v)}t_n),$$

which implies

$$\tau'_n(\kappa_n) = \frac{\kappa_n}{1 + \sqrt{1 + \kappa_n^2}} - \frac{2n}{v\kappa_n}(1 + O(n^{-1}))$$

$$= \sqrt{\frac{n}{n+v}}(1 + h_n)^2(1 + \frac{2n+v}{2(n+v)}t_n)^{-1} - \sqrt{\frac{n}{n+v}}(1 + h_n)^{-2}(1 + O(n^{-1}))$$

$$= \sqrt{\frac{n}{n+v}}\left(2h_n - \frac{2n+v}{2(n+v)}t_n + 2h_n + O(h_n^2)\right)$$

$$= \frac{4\sqrt{n(n+v)}}{2n+v}h_n(1 + O(h_n)). \tag{55}$$

It follows that

$$vk_n\tau'_n(\kappa_n) = 2s_nh_n(1 + O(h_n)). \tag{56}$$

For $0 \leq \zeta \leq r \lesssim (\log s_n/n)^{1/2}$, it is straightforward to see that

$$\sqrt{1 + \kappa_n^2(1 + \zeta)^2} = \sqrt{1 + \kappa_n^2(1 + \frac{2\kappa_n^2(\zeta + O(\zeta^2))}{1 + \kappa_n^2})} = \sqrt{1 + \kappa_n^2}(1 + \frac{\kappa_n^2(\zeta + O(\zeta^2))}{1 + \kappa_n^2})$$

and

$$1 + \sqrt{1 + \kappa_n^2(1 + \zeta)^2} = (1 + \sqrt{1 + \kappa_n^2})(1 + \frac{\kappa_n^2(\zeta + O(\zeta^2))}{(1 + \sqrt{1 + \kappa_n^2})\sqrt{1 + \kappa_n^2}}). \tag{57}$$

Thereby

$$\tau''_n(\kappa_n(1 + \zeta)) = \tau''_n(\kappa_n)(1 + O((\log s_n/n)^{1/2})) = \frac{v}{2n+v}(1 + O(h_n)),$$

which implies

$$\frac{n(n+v)}{v}\tau''_n(\kappa_n(1 + \zeta))r^2 = \frac{s_nr^2}{4}(1 + O(h_n)) = \frac{s_nr^2}{4} + O(\frac{(\log n)^{3/2}}{\sqrt{n}}). \tag{58}$$

Combining (51), (56) and (58), we have

$$w(r) = v\tau_n(\kappa_n) + 2s_nh_nr(1 + O(h_n)) + \frac{s_nr^2}{4} + O(n^{-1/2}(\log n)^{3/2})$$

$$= v\tau_n(\kappa_n) + 2s_nh_nr + \frac{s_nr^2}{4} + o((\log n)^{-1}).$$

Note that

$$w'(r) = v\kappa_n\tau'_n(\kappa_n(1 + r)). \tag{59}$$

Since $r \ll 1$, it follows from (57) that

$$
\tau_n'(\kappa_n(1+r))
$$
$$
= \frac{\kappa_n(1+r)}{1+\sqrt{1+\kappa_n^2(1+r)^2}} - \frac{2n(1+O(n^{-1}))}{v\kappa_n(1+r)}
$$
$$
= \left[\frac{\kappa_n}{1+\sqrt{1+\kappa_n^2}}(1+r)(1-\frac{\kappa_n^2(r+O(r^2))}{\sqrt{1+\kappa_n^2}(1+\sqrt{1+\kappa_n^2})})\right] - \frac{2n}{v\kappa_n}(1-r+O(r^2))
$$
$$
= \left[\frac{\kappa_n}{1+\sqrt{1+\kappa_n^2}}(1+\frac{r}{\sqrt{1+\kappa_n^2}}+O(\frac{2n}{2n+v}r^2))\right] - \frac{2n}{v\kappa_n}(1-r)+O(\sqrt{\frac{n}{n+v}}r^2)
$$
$$
= \left(\frac{\kappa_n}{1+\sqrt{1+\kappa_n^2}} - \frac{2n}{v\kappa_n}\right) + \left(\frac{\kappa_n}{(1+\sqrt{1+\kappa_n^2})\sqrt{1+\kappa_n^2}} + \frac{2n}{v\kappa_n}\right)r + O(\sqrt{\frac{n}{n+v}}r^2).
$$

Now

$$
\frac{\kappa_n}{(1+\sqrt{1+\kappa_n^2})\sqrt{1+\kappa_n^2}} + \frac{2n}{v\kappa_n} = \frac{2\sqrt{n(n+v)}}{2n+v}(1+O(h_n))
$$

and combining this with (55) gives

$$
\tau_n'(\kappa_n(1+r)) = \frac{4\sqrt{n(n+v)}}{2n+v}h_n(1+O(h_n)) + \frac{2\sqrt{n(n+v)}r}{2n+v}(1+O(h_n)) + O(\sqrt{\frac{n}{n+v}}\frac{\log s_n}{n})
$$
$$
= \frac{4\sqrt{n(n+v)}}{2n+v}h_n(1+\frac{r}{2h_n})(1+O(h_n)).
$$

Therefore,

$$
w'(r) = 4v\kappa_n\frac{\sqrt{n(n+v)}}{2n+v}h_n(1+\frac{r}{2h_n})(1+O(h_n)) = 2s_nh_n(1+\frac{r}{2h_n})(1+O(h_n)).
$$

For property (3), the explicit expression for $\tau_n'(\kappa_n)$ is already given by (55). Now consider the explicit estimation of $v\tau_n(\kappa_n)$. Based on the definition of $t_n$ and (54), we have

$$
\tau_n(\kappa_n)
$$
$$
= \sqrt{1+\kappa_n^2} - \log(1+\sqrt{1+\kappa_n^2}) + \frac{1}{4v}\log(1+\kappa_n^2) - \frac{2n+1}{v}\log\kappa_n
$$
$$
= \frac{2n+v}{v}(1+t_n) - \log(1+\frac{2n+v}{v}(1+t_n)) + \frac{1}{2v}\log(\frac{2n+v}{v}(1+t_n))
$$
$$
\quad - \frac{2n+1}{v}\log(\frac{2\sqrt{n(n+v)}}{v}(1+h_n)^2)
$$
$$
= \frac{2n+v}{v}(1+t_n) - \log\frac{2(n+v)}{v} + \frac{1}{2v}\log\frac{2n+v}{v} - \frac{2n+1}{v}\log\frac{2\sqrt{n(n+v)}}{v}
$$
$$
\quad - \log(1+\frac{2n+v}{2(n+v)}t_n) + \frac{1}{2v}\log(1+t_n) - \frac{4n+2}{v}\log(1+h_n),
$$

whence

$$
v\tau_n(\kappa_n) = 2n+v - v\log\frac{2(n+v)}{v} + \frac{1}{2}\log\frac{2n+v}{v} - (2n+1)\log\frac{2\sqrt{n(n+v)}}{v}
$$
$$
\quad + (2n+v)t_n - v\log(1+\frac{2n+v}{2(n+v)}t_n) + \frac{1}{2}\log(1+t_n) - (4n+2)\log(1+h_n).
$$

Since $t_n \lesssim h_n \lesssim \sqrt{\frac{\log s_n}{s_n}} \ll 1$, applying the Taylor expansion yields

$$(2n+v)t_n - v\log(1 + \frac{2n+v}{2(n+v)}t_n) + \frac{1}{2}\log(1+t_n) - (4n+2)\log(1+h_n)$$

$$=(2n+v)t_n - v(\frac{2n+v}{2(n+v)}t_n - \frac{(2n+v)^2}{8(n+v)^2}t_n^2) - 4n(h_n - \frac{h_n^2}{2}) + O(vt_n^3 + nh_n^3)$$

$$=2nt_n + \frac{v^2}{2(n+v)}t_n + \frac{v(2n+v)^2}{8(n+v)^2}t_n^2 - 4nh_n + 2nh_n^2 + O((\log s_n)^{3/2}s_n^{-1/2})).$$

Now $\frac{s_n}{2n+v} = 1 - \frac{v^2}{(2n+v)^2}$ and we have

$$2nt_n - 4nh_n + 2nh_n^2 = -\frac{4nv^2h_n}{(2n+v)^2} + 4nh_n^2 + \frac{2nh_n^2v^2}{(2n+v)^2} - \frac{4nh_n^2v^4}{(2n+v)^4}.$$

From the definition of $t_n$,

$$\frac{v^2t_n}{2n+2v} + \frac{v}{2}\left(\frac{(2n+v)t_n}{2n+2v}\right)^2$$

$$=\frac{4nv^2h_n}{(2n+v)^2} + \frac{2nv^2h_n^2}{(2n+v)^2}\left(1 + \frac{2v^2}{(2n+v)^2}\right) + \frac{8n^2v}{(2n+v)^2}h_n^2 + O((\log s_n)^{3/2}s_n^{-1/2})$$

$$=\frac{4nv^2h_n}{(2n+v)^2} + \frac{2nv^2 + 8n^2v}{(2n+v)^2}h_n^2 + \frac{4nv^4h_n^2}{(2n+v)^2} + O((\log s_n)^{3/2}s_n^{-1/2}).$$

Consequently,

$$(2n+v)t_n - v\log\left(1 + \frac{2n+v}{2(n+v)}t_n\right) + \frac{1}{2}\log(1+t_n) - (4n+2)\log(1+h_n) = 2s_nh_n^2,$$

which yields

$$v\tau_n(\kappa_n) = 2n + v - v\log\frac{2(n+v)}{v} + \frac{1}{2}\log\frac{2n+v}{v} - (2n+1)\log\frac{2\sqrt{n(n+v)}}{v}$$
$$+ 2s_nh_n^2 + o((\log n)^{-1}).$$

The conclusion follows by combining the logarithmic terms.

Recall that $\tau_n$ is strictly convex, so is $w$ and then we see clearly

$$w(r) \geq w(0) + w'(0)r = v\tau_n(\kappa_n) + 2s_nh_nr(1 + O(h_n))$$

and

$$w'(r) \geq w'(0) = v\kappa_n\tau_n'(\kappa_n) \geq 2s_nh_n(1 + O(h_n)).$$

The second part is verified and the proof is completed. $\square$

**Proof of Lemme 2.3.** Recall the definitions

$$\phi(r) = 2r - \left(\frac{2n+v+\frac{1}{2}}{n}\right)\log r, \qquad \beta(r) = \phi((1+r)L_n^2),$$

where $L_n = \left(\frac{n+v}{n}\right)^{1/4}(1+h_n)$.

If $0 \leq v \ll \log n$, then

$$L_n = 1 + h_n + o(n^{-1} \log n).$$

Using the expansion $\log(1 + x) = x - \frac{x^2}{2} + O(x^4)$ for $|x| \ll 1$, we obtain

$$n\phi((1+r)L_n^2) = 2n(1+r)L_n^2 - (2n + v + \tfrac{1}{2})(\log L_n^2 + r - \tfrac{r^2}{2}) + O(n(\log s_n/n)^{3/2})$$

$$= 2n\Big(L_n^2 - 2\log L_n + (L_n^2 - 1)r + \frac{r^2}{2}\Big) + o((\log n)^{-1}).$$

Employing the asymptotic relations

$$\log L_n = h_n - \frac{h_n^2}{2} + o(n^{-1}\log n), \quad L_n^2 = 1 + 2h_n + h_n^2 + o(n^{-1}\log n), \quad L_n^4 = 1 + 4h_n + O(h_n^2),$$

we further simplify to

$$n\phi((1+r)L_n^2) = 2n\Big(1 + 2h_n^2 + 2h_n r + \frac{r^2}{2}\Big) + o((\log n)^{-1}).$$

Similarly,

$$n\phi(L_n^2) = 2n(L_n^2 - 2\log L_n) + O(vh_n) = 2n(1 + 2h_n^2) + o((\log n)^{-1}).$$

Now observe that

$$\partial_x \phi((x^2 + y^2)L_n^2) = 4L_n^2 x\Big(1 - \frac{1 + \frac{2v+1}{4n}}{(x^2 + y^2)L_n^2}\Big) > 0$$

for all $x \geq 1$ and $y \geq 0$, because $L_n^2 - 1 \gtrsim (\frac{\log s_n}{s_n})^{1/2} \gg \frac{v}{n}$. Hence, for fixed $y \geq 0$, the function $\phi((x^2 + y^2)L_n^2)$ attains its minimum at $x = 1$.

Next,

$$\beta''(r) = L_n^4 \, \phi''((1+r)L_n^2) = \frac{2n + v + \frac{1}{2}}{2(1+r)^2} > 0,$$

so $\beta$ is strictly convex on its domain, and consequently $\beta'$ is strictly increasing. The monotonicity gives $\beta'(r) \geq \beta'(0)$ for every $r \geq 0$. Using $L_n^2 = (1 + h_n)^2(1 + O(v/n))$, and $h_n^2 = O(\frac{\log n}{n}) \gg \frac{v}{n}$,

$$\beta'(0) = L_n^2 \phi'(L_n^2) = 2L_n^2 - \Big(2 + \frac{v + \frac{1}{2}}{n}\Big) = 4h_n + 2h_n^2 + o\Big(\frac{\log n}{n}\Big) > 4h_n.$$

Therefore, by the convexity of $\beta$, statement (2) is proved. $\square$

**Proof of Lemma 2.5.** We note that

$$\exp(-c_1 n y^4)\Big(1 + \frac{y^2}{c_2 h_n}\Big)^{-k} \leq 1.$$

Using the elementary inequalities $e^{-x} \geq 1 - x$ and $\log(1 + x) \leq x$, we obtain

$$\exp\Big(-c_1 n y^4 - k\log\big(1 + \frac{y^2}{c_2 h_n}\big)\Big) \geq 1 - c_1 n y^4 - \frac{ky^2}{c_2 h_n}.$$

Consequently,

$$\int_0^{\delta_n} e^{-u_n y^2 - c_1 n y^4}\Big(1 + \frac{y^2}{c_2 h_n}\Big)^{-k} dy \ \le \ \int_0^{\delta_n} e^{-u_n y^2} dy,$$

and

$$\int_0^{\delta_n} e^{-u_n y^2 - c_1 n y^4}\Big(1 + \frac{y^2}{c_2 h_n}\Big)^{-k} dy \ \ge \ \int_0^{\delta_n} e^{-u_n y^2}\Big(1 - c_1 n y^4 - \frac{k y^2}{c_2 h_n}\Big) dy.$$

Performing the substitution $t = \sqrt{u_n}\, y$,

$$\int_0^{\delta_n} e^{-u_n y^2} dy = \frac{1}{\sqrt{u_n}} \int_0^{\sqrt{u_n}\,\delta_n} e^{-t^2} dt = \frac{\sqrt{\pi}}{2\sqrt{u_n}}\, \mathrm{erf}\big(\sqrt{u_n}\,\delta_n\big).$$

The condition $\sqrt{u_n}\,\delta_n \gtrsim \sqrt{\log n} \gg 1$ gives

$$\mathrm{erf}\big(\sqrt{u_n}\,\delta_n\big) = 1 + O\Big(\frac{e^{-u_n \delta_n^2}}{\sqrt{u_n}\,\delta_n}\Big) = 1 + o\big((\log n)^{-1}\big),$$

whence

$$\int_0^{\delta_n} e^{-u_n y^2} dy = \frac{\sqrt{\pi}}{2\sqrt{u_n}}\big(1 + o((\log n)^{-1})\big). \tag{1}$$

Applying the same substitution we also obtain

$$\int_0^{\delta_n} e^{-u_n y^2}\, n y^4\, dy \lesssim \frac{n}{u_n^{5/2}} \int_0^{\sqrt{2u_n}\delta_n} e^{-\frac{x^2}{2}} x^4\, dx \lesssim \frac{1}{\sqrt{u_n}\log n},$$

$$\int_0^{\delta_n} e^{-u_n y^2}\, h_n^{-1} y^2\, dy = \frac{1}{u_n^{3/2} h_n} \int_0^{\sqrt{2u_n}\delta_n} e^{-\frac{x^2}{2}} x^2\, dx \lesssim \frac{1}{\sqrt{u_n}\log n},$$

$$\int_0^{\delta_n} e^{-u_n y^2}\, n h_n^{-1} y^6\, dy \lesssim \frac{n}{u_n^{7/2} h_n} \int_0^{\sqrt{2u_n}\delta_n} e^{-\frac{x^2}{2}} x^6\, dx \ll \frac{1}{\sqrt{u_n}\log n}.$$

Therefore,

$$\int_0^{\delta_n} e^{-u_n y^2 - c_1 n y^4}\big(1 + \tfrac{y^2}{c_2 h_n}\big)^{-k} dy = \frac{\sqrt{\pi}}{2\sqrt{u_n}}\big(1 + O((\log n)^{-1})\big).$$

The proof is completed now. $\qquad\qquad\qquad\qquad\qquad\qquad\qquad\qquad\qquad\qquad\qquad\quad\square$

### Acknowledgment

The authors thank Mr. Xinchen Hu for his helpful discussions and the anonymous referees for their careful reading and constructive comments, both of which have significantly improved this manuscript.

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
