# OpenReview forum: "Quantitative analysis of  rightmost eigenvalue for a large Chiral non-Hermitian random matrix"
_SLADS/Section_A — Accepted by SLADS_Section_A_

### Review · Reviewer_LKga · 2026-05-22

**Summary Of Contributions:**

This paper studies the rightmost eigenvalue of a chiral non-Hermitian Gaussian random matrix. Unlike the Hermitian case where the largest eigenvalue has a Tracy-Widom limit, the non-Hermitian case shows Gumbel behavior at the edge. Previous work only proves the qualitative convergence. Akin to the result on the spectral edge, the current manuscript proves a precise convergence rate of the Berry–Esseen bound for the real part of the rightmost eigenvalue. The authors thus provide another important example (beyond the fundamental Ginibre case) for the quantitative theory of non-Hermitian matrices.

**Audience:**

Yes

**Broader Impact Concerns:**

I do not have any such concern.

**Claims And Evidence:**

Yes

**Requested Changes:**

General comments:
1. It will benefit the readers if, instead of simply saying the correlation structure is different, the authors can provide more detailed comparison of the current paper with previous work on the Ginibre case and on the spectral edge of chiral non-Hermitian matrix. For instance, is the convergence rate faster or slower and are there some explanations/intuitions?
2.  The authors should mention more concretely in page 3 (or other suitable point in the paper), that the essence of the proof of Theorem 1 lies in showing  the $L^2$ norm of restricted kernel converges to 0 at some rate which is faster than the rate of Tr($K_n|_{A(t)}$) approximating $\exp(-t)$. This will also better calrify the individual roles played by the lemmas in Section 2 (instead of just vaguely saying they are related) and make the readers aware of which part provides the dominant contribution to the approximation error.

Other minor comments:
1. Page 4: it would be more helpful to explain a bit more about the $v$-dependence of last term in eq. (2.2), I think it is related to the first product in (2.1), but that does not contain any $v$ factor.
2. Lem 2.4: two `satisfying's in the statement. Also, the formula in the case $v \lesssim \log n$ is inconsistent with the one below (2.6) (which I think is the correct one).
3. Lem 2.6: should $(1+y^2)$ be $(1+y^2)^2$?

**Strengths And Weaknesses:**

The paper is generally well written and clearly organized. The presented result on the convergence rate to Gumbel is very sharp (though maybe not that surprising, given previous series of works on Ginibre). This work enriches the theory on non-Hermitian random matrix and is of considerable interest to the random matrix community. I do not find (significant) weaknesses (other than the few changes indicated in the part below).

---

### Review · Reviewer_cvKh · 2026-06-05

**Summary Of Contributions:**

The paper studies the maximally non-Hermitian chiral Dirac ensemble when $\tau=0$, and proves a sharp Berry--Esseen bound for the normalized rightmost eigenvalue, together with large and moderate deviation principles, when the dimension of the matrix goes to infinity. This work can be viewed as a quantitative refinement of the work of Akemann and Bender, who established the weak convergence of the normalized rightmost eigenvalue. It is also a natural extension of the authors' previous work on the normalized spectral radius of the non-Hermitian chiral ensemble.

**Audience:**

Yes

**Claims And Evidence:**

Yes

**Requested Changes:**

The most serious problem is the proof of Lemma 2.7, where the Laplace estimate (2.20) appears to be incorrect. Indeed, at $L_n^2$ one gets $\phi'(L_n^2)\asymp h_n$ and hence
$\int_{L_n^2}^\infty  e^{-n\phi(x)}dx\lesssim \frac{1}{nh_n}\exp(-n\phi(L_n^2))$.
Consequently, by comparing (2.21) with (2.12) one gets
$\|\mathbb{K}_n|{A(t)}\|_2\lesssim e^{-t}\frac{nh_n^{5/2}}{\sqrt{n}h_n^{3/2}} = e^{-t}\sqrt{n}h_n\lesssim e^{-t}\log n$,
which crucially affects the approximation used in the proof of Theorem 1. This should be able to fixed by a more careful analysis of (2.16), but the current version needs to be revised.

A minor issue is regarding the parameter $v$: in the statement of Theorem 1, the case $v=0$ is allowed,whereas the proofs in Section 2 requires $v\ge 1$. It should not be difficult to extend the current arguments to include $v=0$, but this point needs to be addressed.

Typos and mistakes:
1. The normalizing constants in the joint eigenvalue densities (1.1) and (2.1) are missing.
2. In Lemma 2.3, the derivative statement is a bit off. It seems that $\beta'(r)$ also contains a linear term in $r$, which after factoring out $h_n$ might give an extra $O(1)$ term.
3. In the proof of Lemma 2.3, the endpoint used in Laplace's method is a minimum of the phase, not a maximum.
4. The second bullet of Lemma 2.4 says ``when \(v\lesssim\log n\)'', which seems to be $\log n\lesssim v$.
5. For large $v$ the statement of Lemma 2.4 is fine when $z\asymp w$, but in general the dependence of $w$ is missing.
6. In the proof of the moderate deviation, the error is of order $O(nd_n^3)+O(\log n)$. The assumption $\sqrt{\log n/n}\ll d_n$ does not rule out the possibility that the $O(\log n)$ term is dominant.

Suggestions:
1. $L_n$ itself is a function of $t$, the notation $L_n^2(1+r)$ in the statement of Lemma 2.3 is confusing.
2. The proof of the large $v$ in Lemma 2.7 is omitted with the phrase that ``the same phenomenon persists''. Since this lemma is essential for Theorem 1, the large $v$ case should be written out.
3. The eigenvalue process of non-Hermitian chiral Dirac ensemble should also be determinantal for general $\tau>0$. Does any of the statement can be generalized? This paper studies the rightmost eigenvalue, how about the maximal imaginary part of the eigenvalues?

**Strengths And Weaknesses:**

The overall strategy is natural and appropriate. The proof relies on the fact that the eigenvalues form a determinantal point process with explicit kernel $\mathbb{K}_n$, and the event that the rightmost eigenvalue lies below a threshold is represented as a Fredholm determinant over a certain half-plane. Similarly, the moderate and large deviation principles can be reduced to suitable bounds on the Hilbert--Schmidt norm and trace of the  kernel $\mathbb{K}_n$. The key ingredients are the asymptotic estimates for $\mathbb{K}_n$ in different regimes, established in the lemmas in Section 2. The sharp Berry--Esseen constant is then extracted from a second-order expansion of Tr$(\mathbb{K}_n|{A(t)})$.
This is the correct general approach. However, several parts of the written proof require correction or expansion.
Some issues are typographical or presentational, while others are genuine gaps, see the Requested Changes section below.

---

### Decision · Action_Editor_N6iQ · 2026-06-26

**Recommendation:** Accept as is

**Comment:**

The paper is well written, and the methodology is rigorous. The result enriches the study of non-Hermitian random matrices and is of considerable interest to the random matrix community. Congratulations on a fine piece of work.

**Audience:**

Yes, the findings of this paper would be interesting to at least some people in the journal's audience.

**Claims And Evidence:**

Yes, the claims made in the submission are supported by accurate, convincing, and clear evidence.